EMBO
Molecular Medicine

# Dynamics of multiple resistance mechanisms in plasma DNA during EGFR-targeted therapies in non-small cell lung cancer

Dana Wai Yi Tsui[1,2,§,†] (iD), Muhammed Murtaza[1,2,3,†,¶,††], Alvin Seng Cheong Wong[4], Oscar M Rueda[1,2], Christopher G Smith[1,2], Dineika Chandrananda[1,2], Ross A Soo[4,5], Hong Liang Lim[6], Boon Cher Goh[4,5], Carlos Caldas[1,2,3,7], Tim Forshew[1,2,‡‡], Davina Gale[1,2], Wei Liu[1,2,§§], James Morris[1,2], Francesco Marass[1,2,¶¶,†††], Tim Eisen[3,7,8], Tan Min Chin[4,5,9,‡,*] (iD) & Nitzan Rosenfeld[1,2,‡,**] (iD)

## Abstract

Tumour heterogeneity leads to the development of multiple resistance mechanisms during targeted therapies. Identifying the dominant driver(s) is critical for treatment decision. We studied the relative dynamics of multiple oncogenic drivers in longitudinal plasma of 50 *EGFR*-mutant non-small-cell lung cancer patients receiving gefitinib and hydroxychloroquine. We performed digital PCR and targeted sequencing on samples from all patients and shallow whole-genome sequencing on samples from three patients who underwent histological transformation to small-cell lung cancer. In 43 patients with known EGFR mutations from tumour, we identified them accurately in plasma of 41 patients (95%, 41/43). We also found additional mutations, including *EGFR* T790M (31/50, 62%), *TP53* (23/50, 46%), *PIK3CA* (7/50, 14%) and *PTEN* (4/50, 8%). Patients with both *TP53* and *EGFR* mutations before treatment had worse overall survival than those with only *EGFR*. Patients who progressed without T790M had worse PFS during TKI continuation and developed alternative alterations, including small-cell lung cancer-associated copy number changes and *TP53* mutations, that tracked subsequent treatment responses. Longitudinal plasma analysis can help identify dominant resistance mechanisms, including non-druggable genetic information that may guide clinical management.

**Keywords** circulating tumour DNA; liquid biopsy; lung cancer; resistance mechanisms; targeted therapy
**Subject Categories** Cancer; Pharmacology & Drug Discovery; Respiratory System

## Introduction

Molecularly targeted therapies offer substantial clinical benefit in a subset of patients whose tumours harbour specific oncogenic drivers. Unfortunately, treatment resistance inevitably develops, partly driven by the evolving genetic landscape of cancer cells. For example, though non-small-cell lung cancer (NSCLC) patients carrying activating mutations in *EGFR* (epidermal growth factor receptor) initially respond to EGFR-targeted tyrosine kinase inhibitors (EGFR-TKIs; Lynch *et al*, 2004; Paez *et al*, 2004), the emergence of

1   Cancer Research UK Cambridge Institute, Li Ka Shing Centre, University of Cambridge, Cambridge, UK
2   Cancer Research UK Major Center - Cambridge, Cambridge, UK
3   Department of Oncology, University of Cambridge, Cambridge, UK
4   Department of Haematology-Oncology, National University Cancer Institute, National University Health System, Singapore, Singapore
5   Cancer Science Institute, Centre for Translational Medicine, National University of Singapore, Singapore, Singapore
6   Parkway Cancer Center, Singapore, Singapore
7   Department of Oncology, Addenbrooke's Hospital, Cambridge University Health Partners, Cambridge, UK
8   Oncology Early Clinical Development, AstraZeneca, Cambridge, UK
9   Raffles Cancer Centre, Raffles Hospital, Singapore, Singapore
    *Corresponding author. Tel: +65 63112306; E-mail: csictm@nus.edu.sg
    **Corresponding author. Tel: +44 1223 769769; E-mail: nitzan.rosenfeld@cruk.cam.ac.uk
    †These authors contributed equally to this work
    ‡These authors should be considered as (co-)senior authors and project co-leaders
    §Present address: Department of Pathology, Center for Molecular Oncology, Memorial Sloan Kettering Cancer Center, New York, NY, USA
    ¶Present address: Center for Noninvasive Diagnostics, Translational Genomics Research Institute, Phoenix, AZ, USA
    ††Present address: Mayo Clinic, Center for Individualized Medicine, Scottsdale, AZ, USA
    ‡‡Present address: Inivata Ltd., Granta Park, Cambridge, UK
    §§Present address: University of Glasgow, Glasgow, UK
    ¶¶Present address: Department of Biosystems Science and Engineering, ETH Zurich, Basel, Switzerland
    †††Present address: SIB Swiss Institute of Bioinformatics, Lausanne, Switzerland

mutations that confer resistance to these TKIs or activate alternative drivers (such as EGFR T790M, MET/HER2 amplifications, PIK3CA mutation) leads to eventual drug resistance (Yu *et al*, 2013; Camidge *et al*, 2014). Some of these resistance mechanisms are targetable, such as T790M (Janne *et al*, 2015; Sequist *et al*, 2015) and MET amplification (Sierra & Tsao, 2011). Apart from such individual genetic changes, a small subset of EGFR-mutant NSCLC patients develop resistance to EGFR-TKI therapy by undergoing histological transformation to small-cell lung cancer (SCLC) and become sensitive to standard SCLC treatment (Sequist *et al*, 2011; Niederst *et al*, 2015). Therefore, longitudinal monitoring of the dynamic genetic changes during the course of a patient's treatment has become increasingly important to guide treatment at progression or when resistance occurs. Plasma circulating tumour DNA (ctDNA) is a non-invasive method that has been used to identify EGFR mutations and other genetic drivers in NSCLC and in response to treatment of NSCLC patients with EGFR-TKIs (Yung *et al*, 2009; Couraud *et al*, 2014; Douillard *et al*, 2014; Newman *et al*, 2014, 2016; Weber *et al*, 2014; Paweletz *et al*, 2015; Wan *et al*, 2017). During treatment of NSCLC patients with first-generation EGFR-TKIs, serial assessment of EGFR mutations in plasma ctDNA has proved successful in allowing early detection of T790M-driven resistance prior to radiographic progression (Oxnard *et al*, 2014; Mok *et al*, 2015). However, a subset of the patients develop resistance that is independent of the EGFR pathway, and multiple resistance mechanisms may co-exist because of tumour heterogeneity (Sequist *et al*, 2015; Abbosh *et al*, 2017). Here, we performed longitudinal analysis of plasma ctDNA to study the dynamics of co-existing multiple resistance mechanisms during sequential therapy in NSCLC patients.

In this study, we analysed a cohort of 392 plasma samples collected longitudinally from 50 Stage IV NSCLC patients. All were treated with the first-generation TKI gefitinib in combination with hydroxychloroquine as part of the "Hydroxychloroquine and Gefitinib to Treat Lung Cancer" trial (NCT00809237). Thirty-four patients were TKI-naïve (i.e. not previously treated with EGFR-TKI), and 16 were TKI-treated (i.e. previously treated with TKI with a 2-week washout period). Eligibility for the trial and patient characteristics are summarized in the Appendix Supplementary Methods. This is a phase II study with a phase I lead in that studies the tolerability, safety profile and efficacy of hydroxychloroquine and gefitinib in advanced non-small-cell lung cancer. Appendix Fig S1 summarizes the number of patients in each arm (Appendix Fig S1). We performed tagged-amplicon deep sequencing (TAm-Seq; Forshew *et al*, 2012) for *de novo* identification and quantification of mutations in EGFR exons 18–21, coding regions of TP53 and PTEN, and selected hotspot regions of PIK3CA, KRAS and BRAF; and digital PCR for detection and quantification of hotspot mutations in EGFR. For a subset of patients, we also performed shallow whole-genome sequencing to analyse global copy number changes during treatment (Heitzer *et al*, 2013).

# Results

### Mutational profiling by plasma DNA

To determine whether plasma was a good surrogate of EGFR mutation status in the tumour, we compared the EGFR mutation status in plasma samples (as determined by our assays) with the tumour

status reported in hospital records. The EGFR status was known in the tumour of 43 of the 50 patients, and we detected the same EGFR mutation in any follow-up plasma samples of 41 of 43 (95%) patients (Fig 1A and Appendix Table S1). In the remaining seven patients, two were found to be EGFR wild-type in both tumour and plasma, and the remaining five have EGFR mutations detected in plasma. In 24 patients who responded to the treatment within the initial 70 days, 19 of them showed a drop in EGFR cfDNA levels within that period (Appendix Fig S2 and Appendix Table S2). In addition to EGFR, somatic mutations in other cancer genes, such as in TP53 or the PI3K/AKT/mTOR pathway (PIK3CA and PTEN), were also identified in the plasma of 29 patients (Fig 1B). Of the identified mutations, 25–43% are likely oncogenic (TP53, 10/23, 43%; PIK3CA, 3/7, 43%; PTEN, 1/4, 25%) according to OncoKB annotation (Chakravarty *et al*, 2017). To further compare molecular profiles between tumour and plasma, we studied paired tumour and plasma samples in four patients, where tumour samples were available before and after disease progression. The types of EGFR mutations identified in plasma and tumour (EGFR activating, resistance-conferring mutations in EGFR and other mutations) were identical for 11 of 12 (92%) mutations before treatment, and for 9 of 12 (75%) mutations after treatment (Appendix Fig S3 and Appendix Table S3). Plasma captured the same or more mutations than tumour in 23 of 24 cases (96%). These results confirmed that plasma analysis is informative for mutation profiling in NSCLC patients using our assays. Initial changes in EGFR ctDNA levels after start of treatment mirrored in most cases the radiographic assessment of clinical response.

### Prognostic value of baseline plasma DNA

We studied the relationship between pre-treatment EGFR ctDNA levels and prognosis in 19 TKI-naïve patients (Appendix Table S4), for which at least one plasma sample was collected before initiation of treatment. Patients with low levels of EGFR-activating mutations pre-treatment tended to have better progression-free survival (PFS) and overall survival (OS; Fig 2A and B), though this did not reach statistical significance level of 0.05 (their corresponding Cox *P*-values were 0.06 for both PFS and OS). Of note, patients with low levels of EGFR-activating mutation allele fractions had reduced tumour burden (median 17 mm) by RECIST measurements, as compared to those with intermediate (median 42 mm) and high (median 80 mm) levels of EGFR-activating mutation (Appendix Table S4). These findings suggest that baseline mutation concentrations in the plasma correlate with tumour burden. In addition, patients with both EGFR and TP53 detected in pre-treatment plasma tended to have worse prognosis (Fig 2C and D, Cox *P*-value 0.109 for PFS and 0.035 for OS). We repeated the analysis with copies/ml instead of mutant allele fractions, and the conclusions were the same. These data suggest that both the molecular profile of genomic alterations, and the quantification of ctDNA levels in baseline plasma, can have prognostic implications.

### Mutation dynamics in plasma DNA reveals heterogeneous resistance mechanisms

For 45 of 50 patients, EGFR mutations were detected before treatment in tumour and/or plasma and more than one plasma sample

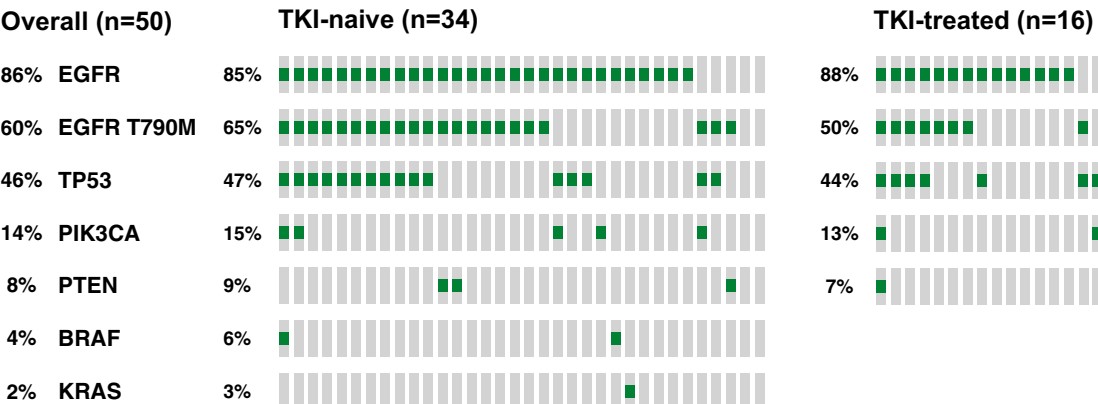

**A**

|  | Total | Ex19del | L858R | Other EGFR | Not detected |
|---|---|---|---|---|---|
| **Tumour** |  |  | Plasma |  |  |
| Ex19del | 23 | 21 | 2* |  | 2 |
| L858R | 15 |  | 15 |  | 0 |
| Other *EGFR* | 3 |  |  | 3 | 0 |
| Not detected | 2 |  |  |  | 2 |

*EGFR* status in tumour vs plasma (including EGFR negative patients): 41/43 (95%)
*Two patients had both EGFR Ex19Del and L858R mutations detected in plasma

**B      Summary of mutations detected in longitudinal plasma of the 50 patients**

| Overall (n=50) | | TKI-naive (n=34) | TKI-treated (n=16) |
|---|---|---|---|
| 86% | **EGFR** | 85% | 88% |
| 60% | **EGFR T790M** | 65% | 50% |
| 46% | **TP53** | 47% | 44% |
| 14% | **PIK3CA** | 15% | 13% |
| 8% | **PTEN** | 9% | 7% |
| 4% | **BRAF** | 6% | |
| 2% | **KRAS** | 3% | |

**Figure 1.  Summary of somatic mutations identified in the 50 NSCLC patients.**

A   Detection of tumour EGFR mutations in plasma. EGFR mutation status in tumour samples was documented in the clinical record for 43 patients (Appendix Table S1), of which 38 had verified hotspot activating mutations (deletion in exon 19 for 23 patients and the L858R mutation for 15 patients), three patients had other mutations in *EGFR* (one of these patients had two different mutations detected in the tumour sample), and two patients were wild-type for *EGFR* according to tumour analysis and confirmed by plasma analysis.

B   Summary of the mutations identified in any of the plasma samples during longitudinal follow-up in the 50 patients. TKI-naïve (*n* = 34) and TKI-treated (*n* = 16) patients are presented separately.

was available from clinical follow-up. Longitudinal analysis of ctDNA in plasma revealed heterogeneity of resistance mechanisms (Fig 3A). During longitudinal follow-up, a large subset of patients retained the sensitizing mutation and developed resistance-conferring *EGFR* T790M mutation (*n* = 28/45, 62%, Fig 3B). To estimate the detection lead time (i.e. the interval between detection of the resistance-conferring mutation in plasma and radiographic evidence of disease progression), we focus on 28 patients where T790M was detected in plasma at any time during EGFR-TKI, including detection before disease progression became evident. In patients treated with first-line EGFR-TKI, we found that the median time-to-appearance of T790M in plasma was 4 months from the start of TKI treatment, with a lead time between T790M detection and clinical progression of 6.8 months. Patients with *EGFR* T790M can now be treated with third-generation, irreversible EGFR-TKIs (Janne *et al*, 2015; Piotrowska *et al*, 2015). One patient (220) had a biopsy of the lung tumour after progression, in which both activating *EGFR* exon 19 and T790M mutations were detected. The same mutations were detected in plasma at the time of progression. This patient was then treated with a third-generation EGFR-TKI (EGF-816, Novartis (NCT02108964)) and demonstrated partial radiological response.

Subsequent plasma samples showed no further EGFR mutations (data shown in Dataset EV1).

In a second group of patients (*n* = 10/45, 22%), the activating *EGFR* mutation was detected in plasma before and after progression, with an average mutant allele fraction (AF, i.e. the fractional concentration of mutant allele over total DNA) of 7.9%, but not T790M (Fig 3C). The continued presence of activating mutations in plasma suggests possible positive selection of the mutations in the *EGFR* pathway in the corresponding cancers. In these patients, mutations in other pathways also emerged in plasma, such as *TP53* and *PIK3CA* (Dataset EV1). One possible hypothesis is that tumours of patients in this group may retain partial sensitivity to EGFR-TKI treatment, and may respond clinically if EGFR-TKI is used in combination with treatments targeting additional resistance pathway.

The third group of patients (*n* = 7/45, 15%) did not have EGFR-activating nor known resistance-conferring mutations in EGFR detected in plasma when they progressed. These patients initially had exon 19 deletion detected in the tumour (7/7) and their first plasma sample (6/7). Interestingly, comparing to the other two groups, this group of patients had EGFR-activating mutations

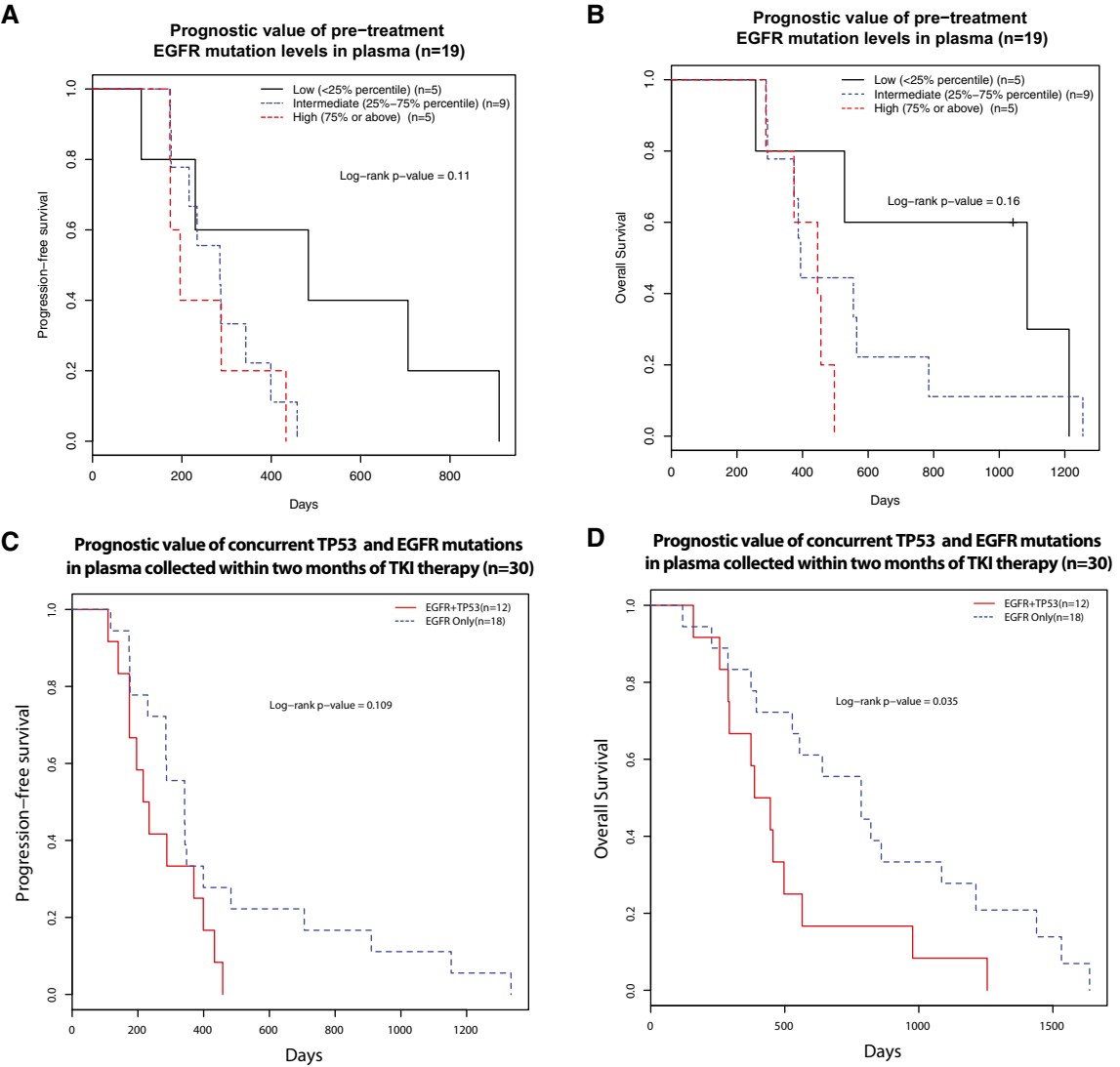

**Figure 2.  Prognostic value of qualitative and quantitative assessments of pre-treatment ctDNA.**

A, B   The relationship of pre-treatment *EGFR*-activating mutation levels (allele fractions) with progression-free survival (PFS) and overall survival (OS) of 19 first-line TKI-treated patients where baseline plasma samples (collected before the start of treatment) were available. Patients were grouped into three groups according to their pre-treatment ctDNA levels, as measured by *EGFR*-activating mutation allele fractions: low (< 25% quartile), intermediate (25–75% quartile) and high (> 75% quartile) ctDNA levels. Kaplan–Meier survival curves indicated that patients with high baseline pre-treatment *EGFR*-activating mutant allele fractions were non-significantly associated with unfavourable (A) PFS (log-rank *P*-value = 0.11) and (B) OS (log-rank *P*-value = 0.16), Cox *P*-value of 0.06 for either PFS or OS.

C, D   The prognostic value of concurrent TP53 and EGFR mutations in pre-treatment plasma samples before EGFR-TKI therapy. This analysis was performed in 30 first-line EGFR-TKI patients where plasma samples were available within 2 months of start of treatment. The presence of both TP53 and EGFR mutations in plasma was associated with a trend of worse PFS (log-rank *P*-value = 0.109, hazard ratio and 95% confidence interval: 0.53 [0.24–1.17]) and significantly worse OS (log-rank *P*-value = 0.035, hazard ratio and 95% confidence interval: 0.43 [0.20–0.97]).

present at relatively lower allele fractions in their first plasma samples [groups 1 and 2: median EGFR mutations mutant allele fractions was 3% (range: 0.07–65.7%) versus group 3: median 0.23% (range: 0.06–2.11%)]. We do not rule out the possibility that the tumours of these patients might release less tumour-derived DNA into the circulation. In some of these patients, we detected alternative cancer mutations such as *TP53* and *PIK3CA* in plasma before treatment was initiated, and the levels of these mutations then increased to present the highest allele fractions in ctDNA when disease progressed (Fig 3D). We speculate that one possible

explanation for the absence of EGFR mutations in cfDNA at disease progression could be that, EGFR mutations were subclonal in those patients initially, and under the selective pressure of the EGFR-targeting therapy, the EGFR-driven clones shrank below detection limit of the assay, while clones that were driven by alterative drivers (such as TP53 and PIK3CA) and did not carry the EGFR-sensitizing mutations, expanded. Based on our data from cfDNA, these alternative drivers pre-existed even before treatment initiation, but were present in parts of the tumour that were not analysed, or alternatively were present at very low cellularity such that

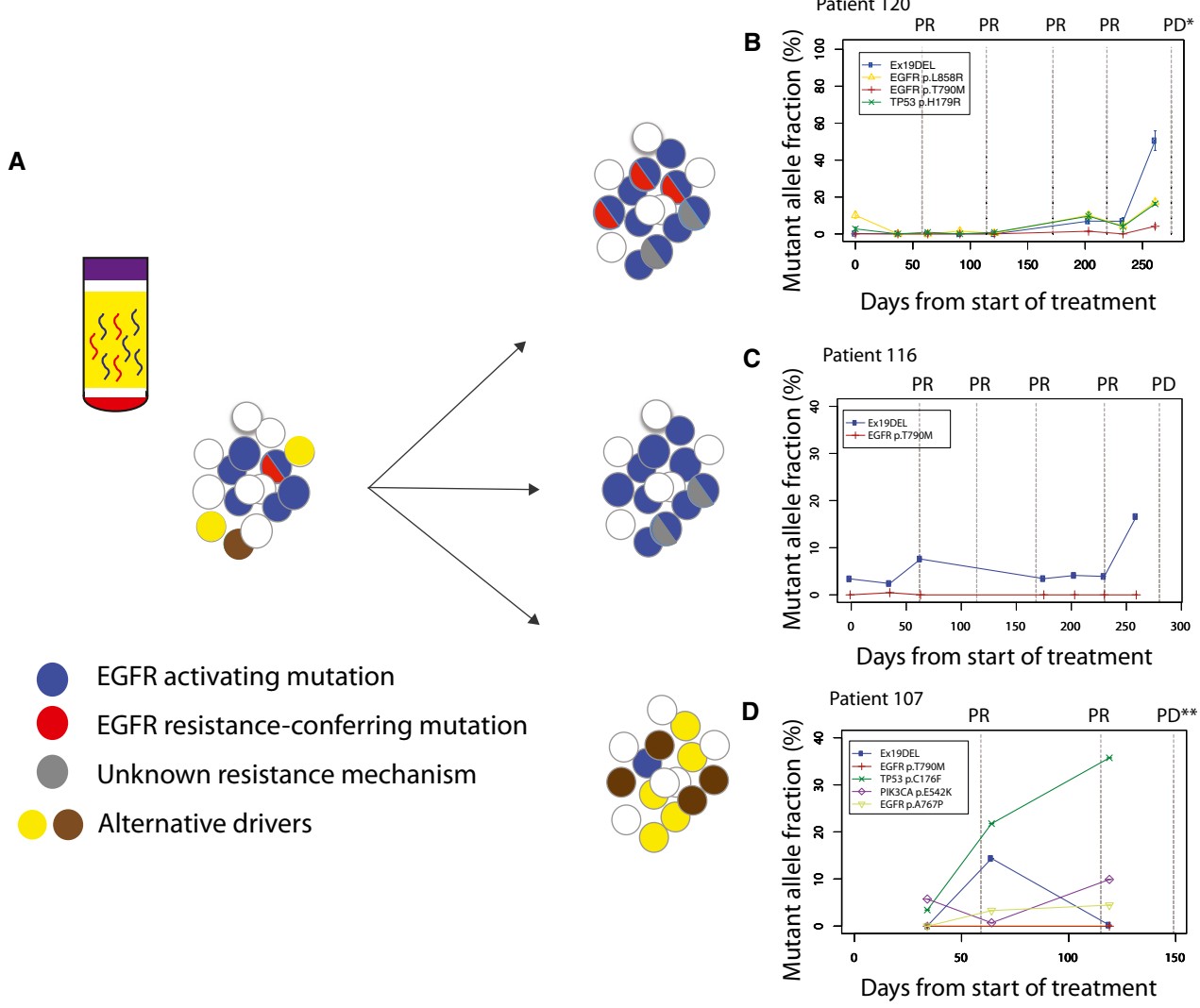

**Figure 3. Longitudinal analysis of ctDNA dynamics reveals distinct patterns of resistance mechanisms.**

A   Longitudinal analysis of ctDNA dynamics in 45 NSCLC patients revealed three main groups of concurrent heterogeneous resistance mechanisms.

B   In the first group (*n* = 28/45, 62%), patients retained EGFR-sensitizing mutations before and after disease progression, with the development of T790M in their plasma samples, indicating that at least some of the progressing clones developed resistance to TKI by acquiring T790M.

C   In the second group (*n* = 10/45, 22%), patients retained EGFR-sensitizing mutations but progressed without developing T790M in their plasma samples, suggesting that resistance arose due to other mechanisms which were not analysed in this dataset.

D   In the third group (*n* = 7/45, 15%), patients progressed without EGFR-sensitizing nor resistance-conferring T790M mutations detected in their plasma samples. Resistance possibly develops through dependence on alternative cancer driver pathways.

Data information: For patients where multiple *EGFR*-activating mutations were identified in plasma, only the most abundant one is shown here (complete data for all patients are shown in Dataset EV1). Clinical progression and CT imaging times are indicated with a dotted line, with RECIST classification: SD, stable disease; PR, partial response; PD, progressive disease. Progressive disease defined by presentation of symptoms on brain or bone scan is indicated by PD**.

their allele fractions in those samples were below the detection limit by standard clinical tumour sequencing assay. Recent data from tumour sequencing suggested EGFR may be subclonal in a small subset of EGFR-mutant NSCLC tumour (McGranahan *et al*, 2015), which agrees with our hypothesis. A recent plasma-based study also reported 4 out of 24 NSCLC patients had EGFR-sensitizing mutations detected in plasma at T0 but absent when the patients progressed, which agreed with our findings (Pecuchet *et al*, 2016). The fact that EGFR T790M was not detected in the third group suggested that the cancers have developed resistance

mechanisms that are alterative to the EGFR pathway, which agreed with the observations of alternative drivers in plasma in some patients. For the patients whom we did not detect T790M nor other drivers were detected, they may have progressed due to other resistance mechanisms that were not covered by our targeted sequencing assay.

Of note, overall, TKI-naïve patients who progressed without T790M detection (13 patients selected from the second and the third groups) had a significantly worse PFS (Appendix Fig S4A and Appendix Table S5, *P*-value = 0.008), and a trend towards worse OS

(Appendix Fig S4B, *P*-value = 0.22), as compared to T790M-positive patients. These observations confirm results from a re-biopsy study (Oxnard *et al*, 2011), suggesting that patients that progress without T790M are less likely to benefit from TKI continuation, possibly due to the more aggressive nature of their disease or reduced dependency on the EGFR pathway.

## Dynamics of EGFR mutations in plasma across multiple lines of treatment

To explore the utility of monitoring both activating and resistance-conferring mutations, we investigated their relative representation in plasma in relation to radiographic assessment in one patient for whom samples were available spanning sequential lines of first-generation EGFR-TKI and chemotherapy (Fig 4). Three lesions were tracked by imaging (Fig 4B): in the lung (L1), left liver lobe (L2) and right liver lobe (L3). During treatment with TKI, all three lesions initially shrank, before a small incremental increase in the size of L3 from day 49, although this did not amount to RECIST progression. The mutant AF of EGFR-activating mutation (exon 19 deletion) decreased initially but increased from day 134 (Fig 4A). The T790M mutation was detected from day 189.

On day 297, L3 showed a substantial growth. The T790M mutation at that timepoint reached AF of 3.6%. Treatment was changed to platinum-based doublet chemotherapy at which point L3 showed

a reduction on serial CT imaging, which coincided with a drop of the T790M mutation AF to 0.7% (Fig 4A). At day 679, despite the shrinkage of L3, L1 and L2 grew, coinciding with an increase in the AF of the activating mutation from 8.6% (when responding to initial TKI) to 43%. The patient was given a first-generation EGFR-TKI re-challenge (newer TKIs were not yet approved at the time this patient was treated), and activating mutation sharply dropped back to 6.4%, corresponding to a reduction in L1 and L2.

From day 217 to 244, both EGFR mutations exhibited a sharp drop in AF in plasma, for reasons we do not understand, and may be related to metabolic effects or technical artefacts. The increased AFs from days 244 to 272 may be a related transient effect. Similarly, the sample collected on day 300 from patient 103 presented an unexpectedly low total cfDNA level (> 10-fold different from the timepoint immediately before and after), which could potentially influence the interpretation of mutant allele fractions at that timepoint. Such variations could be contributed by effects of processing, collection or other technical reasons. We have therefore excluded that timepoint from the analysis. To normalize for these kinds of pre-analytic effects, and to explore the relative representation of the alleles, we calculated the ratio of AFs of the resistance-conferring/ activating EGFR mutations (Fig 4A). The ratio was zero before the start of TKI. It reached a maximal value of 0.43 during first-line TKI (when L3 grew substantially), dropped to 0.02 after chemotherapy (corresponding to a reduction in L3), then rapidly increased upon TKI re-challenge to 0.67.

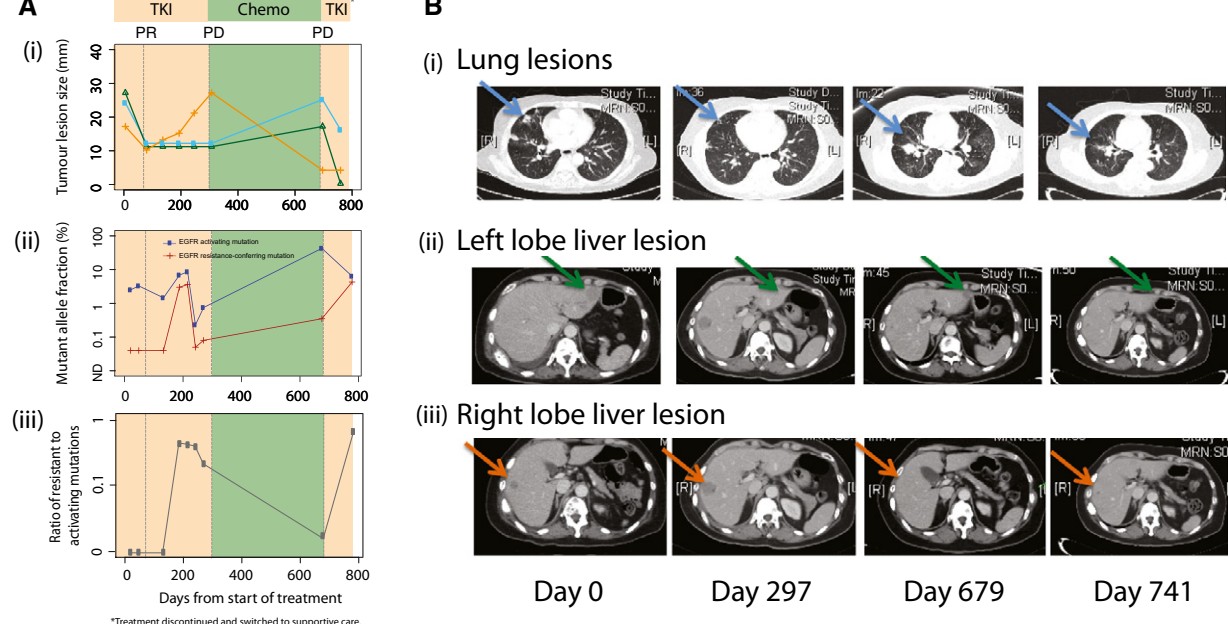

**Figure 4. Analysis of relative representation of activating and resistance-conferring mutations in EGFR during sequential therapy.**

A   (i) Sizes of three different lesions in patient 103 over time, measured from the start of first-line TKI treatment. Shading indicates duration of treatment with TKI (days 0–297), chemotherapy (days 297–679), and TKI re-challenge (days 679–783). From day 783, the patient was treated with supportive care. Dotted lines indicate the CT imaging assessment at select timepoints. (ii) Levels of activating EGFR mutations (exon 19 deletion) and resistance-conferring EGFR T790M mutations for patient 103 (Dataset EV1). (iii) Ratio of resistance-conferring/activating mutations, calculated from data shown in Appendix Table S1 (excluding the data at *T* = 300 days).

B   CT imaging scans performed at the start of TKI treatment (day 0), at the change of treatment to chemotherapy (day 297), at the end of chemotherapy and start of TKI re-challenge (day 679), and after initiating TKI re-challenge (day 741). Sizes are assessed from CT imaging scans, and indicated by blue (i, lung), green (ii, left lobe liver) and orange (iii, right lobe liver) lines. Lesions identified in the lung (blue arrow), left lobe liver (green arrow) and right lobe liver (orange arrow) are indicated. PR, partial responses; PD, progressive disease.

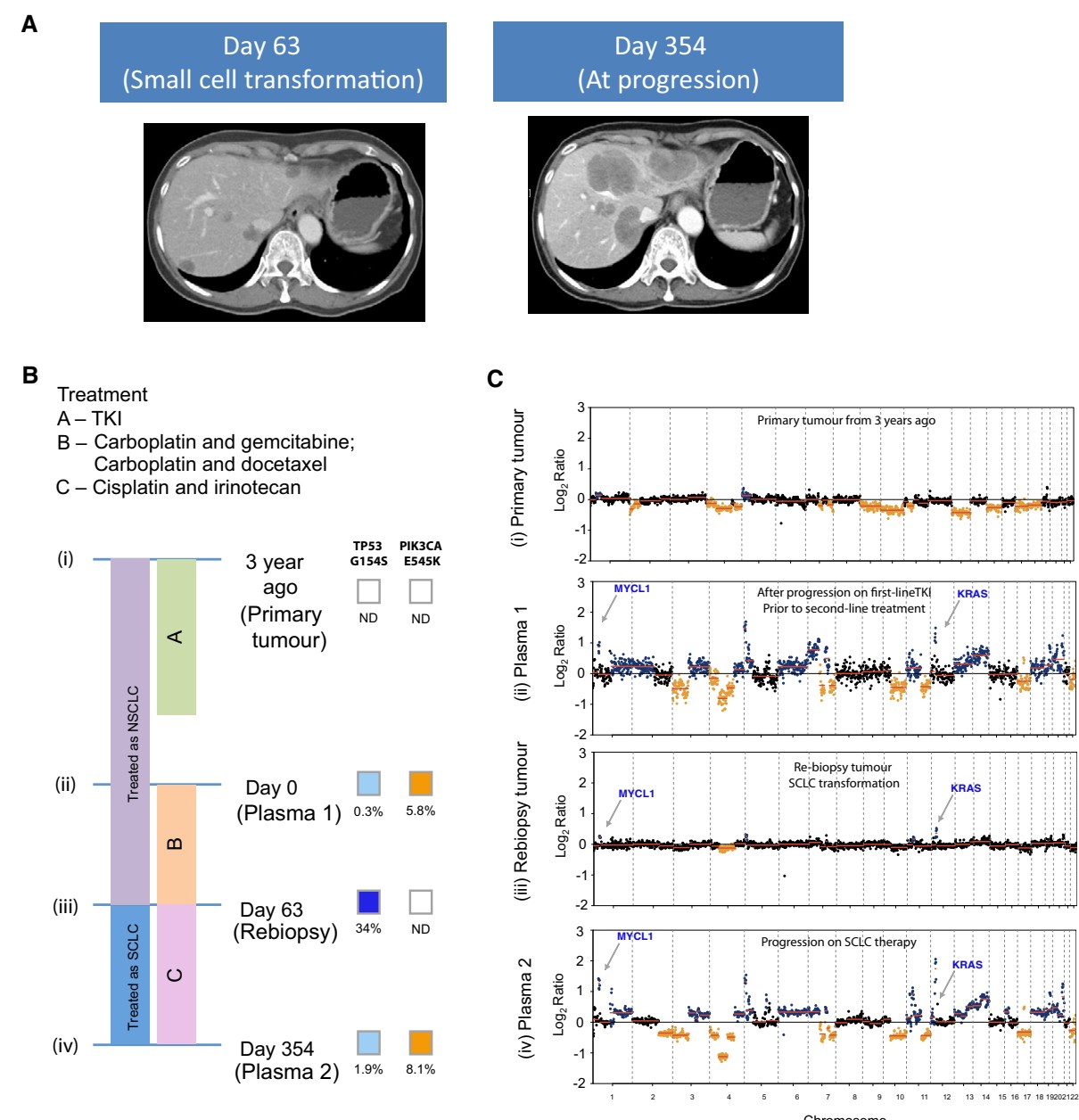

**Figure 5. Plasma analysis reveals global copy number changes and ctDNA dynamics in patients who have undergone histological transformation to SCLC (patient 223).**

A   CT liver scans are shown: At day 63, the patient had progressed on platinum-based chemotherapy, and CT of the liver showed appearance of new liver lesions. Liver biopsy at that point confirmed small-cell lung cancer in new lesions in the liver. CT of the liver at day 354 shows marked growth in the liver lesions, after a period of transient response to cisplatin and irinotecan.

B   The timeline of the patient's treatment is charted, alongside the timepoints where plasma and tumour samples are available for molecular analyses. Per diagram timeline, tumour samples were available from diagnosis (3 years prior to recruitment for study) (i) and at day 63 (iii). Plasma samples were available at day 0 (ii) and day 354 (iv). The mutations allele fractions for TP53 and PIK3CA are shown.

C   Global copy number profiles in plasma samples collected prior to small-cell transformation (ii) and after SCLC transformation and progression on cisplatin and irinotecan (iv). Global copy number profiles in tumour samples collected at diagnosis of NSCLC (i) and at small-cell transformation (iii). CNA events that were significantly identified and coincide with literature-reported SCLC events are denoted in colours: blue for gain and orange for loss.

Although a liver biopsy was not performed, which may have been able to confirm the presence of T790M in L3, the clinical and radiological evidence in conjunction with the dynamics of EGFR mutations in plasma strongly suggests that the T790M was present in L3 but not in L1/L2. This suggests that the ratio of resistance-conferring/activating mutations in plasma can help identify dominant drivers of disease and progression in real time (Oxnard *et al*, 2016).

## Copy number changes and mutations detected in plasma after SCLC transformation

Recent findings have shown that around 2–3% of NSCLC patients develop resistance to EGFR-targeted therapies by undergoing histological transformation to small-cell lung cancer (SCLC; Sequist et al, 2011). In our cohort, three patients (patients 122, 223, 218) were confirmed to present SCLC histology based on re-biopsy examination at progression. We used targeted deep sequencing and shallow whole-genome sequencing of plasma DNA to track and study the dynamics of somatic point mutations and global copy number alterations (CNAs) in samples collected before and after SCLC transformation in those patients.

T790M mutations were not detected in plasma at disease progression for any of the three SCLC-transformed patients, and two of them (patients 122 and 218) retained the EGFR-activating mutations in plasma after SCLC transformation. TP53 mutations were detected before EGFR-TKI initiation in all three patients' baseline plasma samples, at low levels (< 1%) compared to the EGFR-activating mutations, and their levels in plasma increased with disease progression (patient 122, 223) or decreased when patient demonstrated clinical response (patient 218) during small-cell lung cancer-directed chemotherapy (Figs 5 and 6, and Appendix Fig S5). Analysis of the plasma samples collected after transformation in all three patients revealed the emergence of CNAs that have been previously reported to be associated with SCLC, including MYCL1, SOX2 and SOX4 (George et al, 2015; Figs 5 and 6, and Appendix Fig S5). In each patient, we also identified gain or loss of cancer genes as part of larger chromosomal events (> 5 Mb; Figs 5 and 6, and Appendix Fig S5). These may have contributed to the biological change or may represent passenger events as a result of greater genomic instability of the TP53-mutant clones. We observed focal copy number changes at key oncogenic drivers that may play a role in driving disease progression, for example, amplification of KRAS in patient 223 (Fig 5) and amplification of EGFR in patients 122 and 218 (Appendix Fig S5 and Fig 6).

Patient 223 (Fig 5A) initially harboured an exon 19 activating EGFR mutation, had indolent disease and remained clinically and radiologically responsive to first-line EGFR-TKI for 2 years. Subsequently, there was development of PET FDG-avid, but subcentimetre, liver lesions that were not clearly appreciated on CT imaging. At this point, even though progression was only obvious in two new small spots on PET imaging (stable disease by CT and RECIST criteria), ctDNA showed multiple copy number changes (Fig 5B and C, plasma 1). There was subsequent rapid growth of the liver lesions,

despite two lines of chemotherapy. A liver biopsy was performed at that point, and showed only small-cell carcinoma, with no further activating nor T790M EGFR mutations found in the small-cell cancer. Figure 5C demonstrates that a few months prior to the confirmation of small-cell transformation, plasma analysis already showed marked copy number changes, this time of focal amplification of MYCL1 and KRAS, known oncogenic drivers in the KRAS pathway. These focal genomic changes were also observed in the subsequent liver biopsy, showing parallel changes in both tumour and plasma, suggesting that plasma is a good surrogate for study of genomic copy number alterations and evolution. In this particular case, the copy number changes were more markedly observed in the plasma, compared to the tumour, likely due to scarcity of tumour cells in the repeat biopsy.

Another patient with SCLC transformation (patient 218, Fig 6A) harboured an EGFR L858R-activating mutation at diagnosis, and again demonstrated a good clinical and radiological response, with progression-free survival of 14 months on first-line EGFR-TKI (Fig 6B). Upon clinical progression while on treatment with EGFR-TKI (day 78), the plasma analysis (Fig 6B plasma 1) showed increased levels of EGFR L858R as well as a TP53 R175H mutation, one of the most frequently observed mutations in TP53. On treatment with platinum-based doublet chemotherapy with pemetrexed, the patient achieved stability of disease. There was however early progression on maintenance pemetrexed and despite a switch to carboplatin and docetaxel, the patient's lung mass progressed, and a biopsy confirmed SCLC, harbouring the original activating EGFR L858R mutation. At this point, the patient was switched to treatment for SCLC with carboplatin and etoposide, with a radiological response and corresponding reduction in TP53 R175H mutation (Fig 6B, plasma 3). Unfortunately, the treatment response was transient, and there was development of widespread symptomatic metastases in both the brain and spine, necessitating radiotherapy to those areas. At the completion of radiotherapy, the patient had developed liver metastases with rapid progression. At this point, we observed marked copy number change in plasma 4 and 5 analyses (Fig 6C and Appendix Fig S6). This example illustrates that dynamics of genomic copy number changes in plasma reflect the mutational burden and radiologic responses.

The losses of TP53 and RB1 are common in SCLC (George et al, 2015), and we therefore attempted to look for RB1 somatic copy number alterations but did not find any significant signal by sWGS. In patient 218, RB1 did appear to have a reduced copy number, but this event could be part of a much larger chromosomal aberration, and thus, we cannot rule this out as being a passenger event. RB1

**Figure 6. Plasma analysis reveals global copy number changes and ctDNA dynamics in patients who have undergone histological transformation to SCLC (patient 218).**

A  CT images of the left lung tumour at baseline (day −730), and upon progression on EGFR-TKI (day 78). The patient was treated for non-small-cell lung cancer from days 78 to 329 with two lines of chemotherapy: initially carboplatin and pemetrexed, followed by docetaxel. The CT image corresponding to plasma 2 and re-biopsy of the tumour at day 329 was upon progression on the above two lines of chemotherapy. CT image corresponding to plasma 3 at day 379 was upon response to small-cell lung cancer chemotherapy. CT corresponding to days 500 and 524 are upon progression on cisplatin and etoposide. Marked growth of the lung and liver lesions are demonstrated, and this corresponds to marked CNA changes on plasma drawn on those respective days.

B  The timeline of the patient's treatment is shown, alongside timepoints where tumour and plasma samples were available for analyses. The bottom chart shows the respective mutations that were found, and the changes to the mutation allele fractions in a longitudinal timeline.

C  Global copy number profiles in tumour and plasma samples are shown: Tumour samples at baseline diagnosis of non-small-cell lung cancer (day −730) and at transformation to small-cell lung cancer (day 329). For plasma samples, the following were available: days 78 (upon progression on EGFR-TKI); 329 (at transformation to small-cell lung cancer); 379 (at response to small-cell lung cancer); 500 and 534 (progression on small-cell lung cancer treatment).

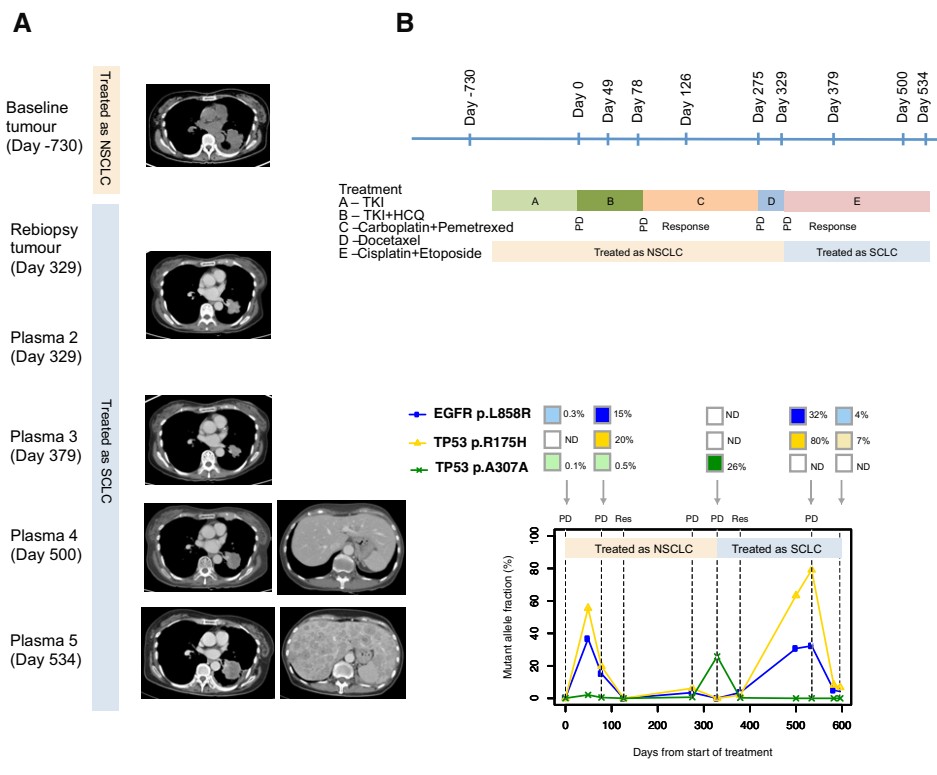

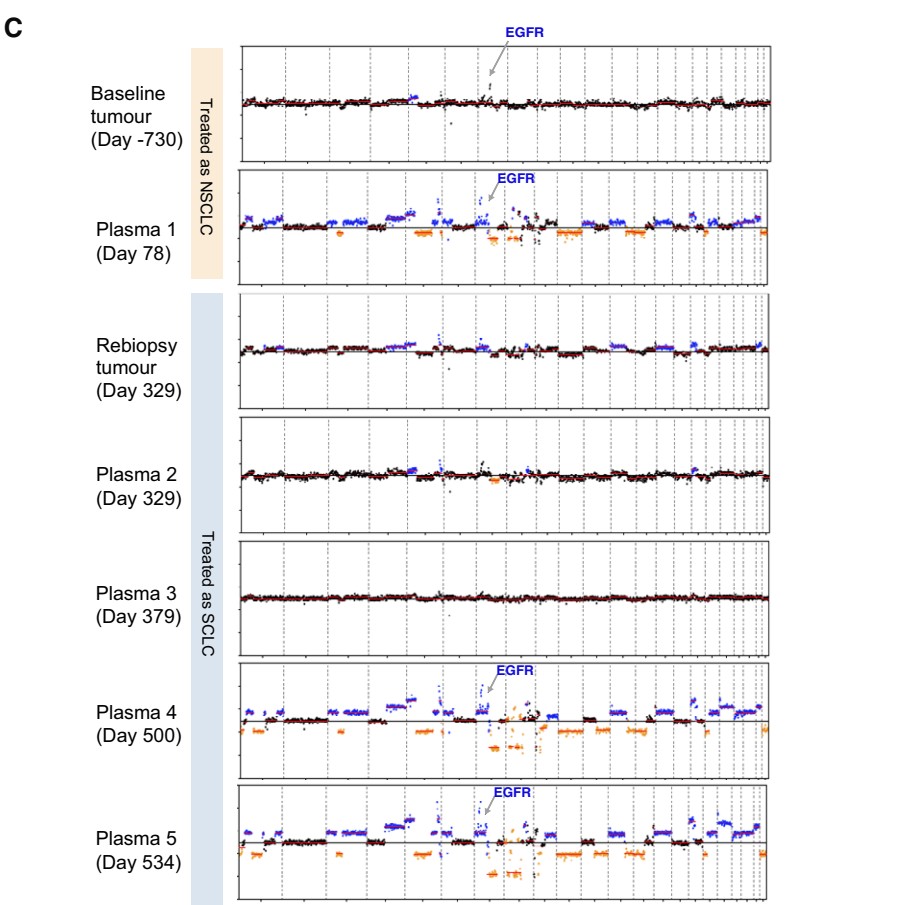

**Figure 6.**

alterations are important driver in SCLC, but it is not necessarily the only driver (Karachaliou *et al*, 2016). All three SCLC-transformed patients have evidence of *TP53* mutations in their SCLC re-biopsies, and pre- and post-transformation plasma suggest that *TP53* is an important driver in these particular patients.

In all three patients, CNAs were more evident in plasma as compared to tumour DNA analysis, likely due to scarcity of tumour cells at repeat biopsy. The data also illustrate that the global CNA profile in plasma can act as an indicator of disease burden and can be used to track clinical progression, as previously suggested in other cancer types (Heitzer *et al*, 2013). Subtype-specific mutational and CNA signatures can be identified in plasma in association with histological transformation that warrants different treatment strategy, and their increasing levels in plasma can pre-date radiological progression (by CT imaging). These observations suggest that plasma genomic changes could be indicative of early progression, and may complement current imaging modalities in monitoring response.

## Discussion

We studied the dynamics of concurrent somatic point mutations and copy number alterations in plasma DNA during treatment of NSCLC patients with EGFR inhibitor. Several observations may provide information for the design of future plasma DNA studies of patients treated with targeted therapies:

First, we found a strong concordance between *EGFR* status in tumour and plasma samples, and showed that mutations in multiple cancer-related genes can be identified directly in plasma by targeted sequencing. These results confirm findings from a previous validation study (Douillard *et al*, 2014; Weber *et al*, 2014; Huang *et al*, 2017), and lend further credibility to the application of circulating DNA in plasma for non-invasive molecular profiling and treatment stratification (Jamal-Hanjani *et al*, 2017; Remon *et al*, 2017). Second, high pre-treatment levels of ctDNA, and specifically of *EGFR* mutations in plasma prior to treatment with EGFR-TKI, correlated with increased tumour burden and were associated with poor prognosis, echoing previous findings (Mok *et al*, 2015). We also showed that early changes in levels of ctDNA (in our case, of *EGFR* mutations) may predict initial response (Parkinson *et al*, 2016). Both of these findings lend support to the analysis of baseline and subsequent plasma samples for *EGFR* mutations to track treatment responses. Third, we detected the emergence of the T790M mutation in approximately 50% of patients who progressed on TKI, at a median of 6.8 months before clinical progression. Early identification of emerging resistance highlights the potential to use ctDNA to guide clinical interventions such as therapies that target T790M-mutant cells (Janne *et al*, 2015; Sequist *et al*, 2015; Chabon *et al*, 2016; Remon *et al*, 2017).

In addition, we showed that profiling TP53 in plasma before EGFR-targeted therapy can provide prognostic value. Cancers harbouring both *TP53* and *EGFR* mutations in baseline plasma were associated with inferior overall survival in patients treated with EGFR-targeted TKI. This confirmed observations from a tumour sequencing study (Labbe *et al*, 2017), and because plasma DNA captures the mutations coming from different parts of the tumour, we found that in some patients, these TP53 mutations pre-existed at

AF < 1% in plasma prior to treatment and later became the dominant mutations in plasma when patients progressed or exhibited histological transformation to SCLC. This echoes one of the recent findings that EGFR-TKI-resistant SCLCs can branch out from early events that pre-existed in NSCLC prior to transformation based on tumour biopsy analysis (Lee *et al*, 2017).

These data highlight the potential value for clinical management of analysing mutations which may not be perceived as "actionable", such as mutations in the tumour suppressor *TP53*, and suggests that the genetic context of an EGFR-mutant tumour may determine its dependence on the EGFR and/or other pathways and predict sensitivity or resistance to EGFR-directed treatment. Tracking the dynamics of multiple mutations showed that different resistance mechanisms co-existed, and are likely to be the result of tumour heterogeneity (Piotrowska *et al*, 2015). As illustrated by the above clinical cases, the response and progression of different lesions coincided with the changing levels of distinct mutations in plasma. As treatment selection pressure is dynamic, analysing the relative proportions of different oncogenic drivers in plasma at any one point may provide insight on a particular dominant oncogenic pathway dependence and guide decisions on subsequent treatment by targeting the current dominant clone.

Finally, we showed that tracking the dynamics of plasma EGFR mutations alone may not provide the most accurate estimate of tumour responses, as seen in the 14% of patients who progressed with decreasing levels of EGFR mutations in plasma. We speculate that this observation may be explained by the recent finding of subclonal EGFR driver mutations in 3 of 21 (14%) NSCLC cases (McGranahan *et al*, 2015), and suggest that monitoring EGFR-targeted therapies by ctDNA would require tracking mutations beyond EGFR. These findings will need to be confirmed by further studies in larger cohorts. Nonetheless, using a multi-gene assay, our data revealed the presence of concurrent oncogenic drivers before treatment. Parallel analysis of somatic point mutations and global CNA events (by shallow WGS) could more accurately track disease burden and detect subtype-specific events (e.g. marked copy number changes associated with histological transformation). In particular, we identified multiple genomic changes in the plasma of three patients who underwent transformation to SCLC (patients 122, 223 and 218), which correlated closely with burden of disease. All three patients presented increasing fractions of *TP53* mutations that pre-existed at < 1% levels in plasma before treatment, together with recurrent SCLC CNA events in ctDNA at the time of transformation. These changes correlated closely with burden of disease, and CNA signals in plasma reduced accordingly when computerized tomography imaging showed radiological response to chemotherapy. In these cases, it was interesting to note that all cases exhibited different drivers, *TP53* in the first case, *MYCL-1* and *KRAS* in the second, and two *TP53* mutations in the third. It may be that each patient has a unique signature of genomic copy number change that may reflect disease burden, and this can be used to determine responding or progressive disease at various timepoints, with relation to each line of treatment received. However, the complex and varied genomic landscape in these transformed small-cell lung cancers underscores the difficulty in targeting any one of these genomic signatures (even if actionable), and providing support for the use of chemotherapy, which is currently the most appropriate treatment for targeting these multiple genomic instabilities. Our

findings suggest that an observation of multiple genomic copy number changes in the plasma of a patient with rapid progression of disease on EGFR-TKI should prompt the need to re-biopsy to exclude the possibility of small-cell transformation. To date, EGFR T790M remains the main actionable resistance mechanisms in the context of EGFR-TKI. However, the ability to reveal TP53 mutations or other possible SCLC-associated genomic signatures in plasma would provide additional insight into possible resistance mechanisms that are particularly important in individuals that show no evidence of T790M or other known resistance mechanisms, and may justify the need for a re-biopsy to confirm the histological transformation.

Our study, spanning 392 clinical samples analysed by a combination of genomic techniques, describes multiple genomic changes in *EGFR*-mutant patients with acquired resistance to first-generation EGFR-TKIs, and may explain the heterogeneity of treatment response to EGFR-TKIs in *EGFR*-mutant patients with similar activating mutations. We studied key genomic driver events of lung cancer, and evaluated their significance in a temporal and dynamic way with respect to disease response and progression in patients with analyses of longitudinal plasma studies. As the majority of patients were treated when second- and third-generation EGFR-TKIs were not readily available, most patients with acquired *EGFR* T790M mutation were not routinely re-biopsied and were treated with chemotherapy. Analyses of changes in ctDNA in response to treatment with second- and third-generation EGFR-TKIs are not within the scope of this study. Nonetheless, some of the acquired resistance mechanisms described here may also apply to acquired resistance to a wide range of EGFR-TKIs. Importantly, our study represents the first report on ctDNA changes in EGFR-mutant cancers before and after histological transformation to SCLC and provides important insight into the management of this alterative form of resistance mechanism to EGFR-TKI.

In summary, our data show that in the NSCLC EGFR-targeted therapy setting, analysing the presence and dynamics of both actionable oncogenic drivers (such as *EGFR* mutations) and other, potentially "non-actionable" alterations (such as *TP53* mutations and global copy number changes), before and during treatment, can offer clinically relevant information to potentially guide subsequent clinical management.

# Materials and Methods

### Sample collection and processing

Patients with metastatic NSCLC treated by gefitinib in combination with hydroxychloroquine therapy attending the University Hospital of Singapore, Singapore, during January 2009 to May 2014 were recruited as part of the "Hydroxychloroquine and Gefitinib to Treat Lung Cancer" study (NCT00809237). This was a single-arm phase II study that recruited two groups of patients. In the first group, EGFR-TKI-naive patients who were known to have activating *EGFR* mutations were recruited to determine whether hydroxychloroquine improved the efficacy of gefitinib. The second group included patients who had previously responded to EGFR-TKIs for at least 12 weeks (per Jackman criteria for acquired resistance to EGFR-TKI) and aimed to determine whether the addition of hydroxychloroquine to gefitinib would reverse acquired resistance in these

patients. Please see consort diagram (Appendix Fig S1) for further details including number of patients in each arm included for the purpose of this cDNA study (note that this represents a subset of the original clinical study—as this was subject to availability of plasma samples). Blood was collected from patients in a CPT sodium citrate tube (BD) every 4 weeks and stored at −80°C. CT imaging of relevant measurable sites of disease was performed every 8 weeks. This study was approved by the Singapore National Healthcare Group (Singapore National Healthcare Group Domain Specific Review Board NHG DSRB Reference: 2008/00196). Blood and tumor collection were also collected and approved by NHG DSRB 2014/00131. Informed consent has been obtained from all patients involved in this study. DNA was extracted from 0.8 to 2 ml of plasma using the Qiagen QIAamp Circulating Nucleic Acid Kit (Qiagen) and eluted into 50 μl buffer AVE. More details of sample processing are given in Appendix Supplementary Methods. A spike-in control, non-human DNA PCR product was added to the lysis buffer during DNA extraction to control for extraction efficiency.

### Mutation identification and quantification by TAm-Seq and digital PCR

Analysis by tagged-amplicon deep sequencing (TAm-Seq) was performed using the panel described previously (Forshew et al, 2012), with the addition of an amplicon that covers exon 18 of *EGFR* (additional details in Appendix Supplementary Methods). All samples were analysed by at least two replicates to control for errors arising during PCR. The purified libraries were sequenced using paired-end 100 bp read length of a HiSeq 2000 or HiSeq 2500 System (Illumina, USA). Somatic mutations were identified based on filtering against the matched normal control (white blood cells) of the same patient. For quantification of known hotspot mutations in EGFR, namely exon 19 deletion, T790M and L858R, digital PCR analysis was developed and optimized: sensitivity and specificity and limit of detection was determined using samples with known mutations and samples from healthy volunteer controls (Appendix Fig S7 and Appendix Table S6). Assays were performed using the BioMark system using 12.765 Digital Arrays (Fluidigm, USA) following manufacturer's instructions and protocol. Total DNA levels (amplifiable copies per ml) were also quantified in every plasma sample by digital PCR using a 65-bp assay targeting a region on the *RPP30* gene (Appendix Table S7), a region in the genome that is not amplified in lung cancer (Wang et al, 2010). Two samples were excluded from analysis due to an unexplained sharp drop in total circulating DNA levels extracted from plasma, with >10-fold drop in those levels compared to samples collected few weeks prior and no evident association with any clinical or treatment parameters. This is suspected to be related to technical fault either at collection or processing of samples. Evaluation of the specificity and sensitivity of the assays was described in Appendix Supplementary Methods. The primer and probe sequences of all the digital PCR assays are summarized in Appendix Table S7. The mutant allele fractions measured by TAM-Seq and ddPCR strongly correlate with each other (Appendix Fig S8).

### Shallow whole-genome sequencing (sWGS)

Libraries were prepared from either plasma DNA (5–10 ng), sheared tumour DNA, or sheared buffy coat DNA using the Plasma-Seq

protocol (Rubicon, USA). Briefly, end repair and "A-tailing" of fragment ends preceded the ligation of truncated Illumina sequencer-compatible adapters to fragment ends. Thermocycling of libraries completed the adapters through the addition of sample-specific index sequences, and was performed as described in the Plasma-Seq protocol (Heitzer *et al*, 2013), using 10 (plasma) or 8 (tumour and buffy coat) amplification cycles. Upon amplification libraries were cleaned using Agencourt AMPure XP beads (Beckman Coulter, USA) at a 1:1 (v/v) ratio and eluted in 30 μl nuclease-free water. Successful library preparation was confirmed by running 1 μl of library on a High-Sensitivity Bioanalyser gel, and libraries were quantified using SYBR-green-based qPCR (Kapa Biosystems, USA). Libraries were pooled in an equimolar fashion, and 125-bp paired-end sequencing was performed on Illumina sequencers (Illumina, USA).

Paired-end sequence reads were aligned to the human reference genome (GRCh37) using BWA, SAMtools was used to convert files to BAM format, to which mate pair information was added. PCR duplicates were marked using Picard-Tools' "MarkDuplicates" feature and were excluded from downstream analysis. Fragment lengths were analysed using Picard-Tools' "CollectInsertSizeMetrics". CNA calling was performed in R using the program QDNAseq (Scheinin *et al*, 2014). Briefly, sequence reads were allocated into equally sized (here 1 Mb and 50-kb bins) non-overlapping bins throughout the length of the genome. Read counts in each bin were corrected to account for sequence GC content and mappability, and regions corresponding to previously "blacklisted" regions (ENCODE) were excluded from downstream analysis. Within the QDNAseq package, bins were segmented using the "Circular Binary Segmentation" algorithm (Venkatraman & Olshen, 2007) and significantly "amplified" or "lost" regions were called using CGHcall (van de Wiel *et al*, 2007).

### Data deposition

Sequence data have been deposited at the European Genome-phenome Archive (EGA), which is hosted by the EBI and the CRG, under Accession no. EGAS00001002908.

### Survival analysis

Kaplan–Meier curves were computed for prognostic groups defined by their mutation fractions, and log-rank tests were computed for testing differences in survival. We measured pre-treatment ctDNA levels using allele fractions of the *EGFR*-activating mutations and computed Kaplan–Meier survival curves to evaluate the effects of different levels of pre-treatment ctDNA: We divided patients into three groups: low pre-treatment ctDNA levels (less than the lower quartile), intermediate (second and third quartiles) or high (upper quartile). All survival analyses were performed using the R package survival (Therneau, 2012). It should be noted that only the EGFR-TKI-naïve group of patients with available pre-treatment plasma samples ($n = 19$) were used for progression-free and overall survival analyses, and correlative prognostic study. This is to ensure analyses of a homogeneous population. The survival analyses of both groups of patients will be reported in a separate clinical paper that would include response rates and other clinical parameters.

The experiments conformed to the principles set out in the WMA Declaration of Helsinki and the Department of Health and Human Services Belmont Report.

### The paper explained

#### Problem
The cancer genome evolves under the selective pressure of targeted therapies. One of the key challenges is to identify resistance mechanisms and the most dominant drivers as early as possible. Analysis of plasma cell-free DNA allows one to track molecular dynamics non-invasively.

#### Results
This study found that cell-free DNA analysis reveals clinically important information during EGFR-targeted therapy in non-small-cell lung cancer (NSCLC): At baseline, quantitative tumour-derived cell-free DNA levels in plasma provided prognostic information and correlated with tumour burden. During treatment, multiple potential indications of resistance such as *EGFR* T790M or *TP53* can be detected in plasma months before disease progression became clinically evident. Longitudinal analysis of tumour-derived cell-free DNA levels tracks tumour responses and reveals heterogeneous resistance mechanisms: The majority depend on the EGFR pathway while a small subset developed alterative drivers that could be identified by tracking multiple mutations in plasma DNA. In patients who developed resistance by transforming to small-cell lung cancer (SCLC), we identified *TP53* mutations, one of the key drivers of SCLC, and SCLC-specific copy number events in plasma before the transformation.

#### Impact
Parallel analysis of multiple mutations and copy number alterations in plasma allows identification of dominant drivers at any given time during treatment. Tracking EGFR mutations alone in plasma during EGFR-targeted therapy may not accurately reflect tumour burden due to underlying tumour heterogeneity. The results of this study provide important insight about the implication of cell-free DNA analysis for management of targeted therapies.

Expanded View for this article is available online.

## Acknowledgements

We are grateful to the following physicians who recruited patients on the study—Dr. Pei Jye VOON, Dr. Winnie LING, Dr. Thomas SOH, Dr. Chee Seng TAN, Dr. Angela PANG and Dr. Yew Oo TAN. We also thank the following for technical support—Maricel Tiemsin CODERO, Hui Hui SHEE, Shiau Hui Diong, and Dr. Azhar ALI from The Centre for Translational Research and Diagnostics, National University of Singapore and Cancer Science Institute, Singapore. We thank the sequencing support by the Genomics Core of the Cancer Research UK Cambridge Institute, and by David Bentley and Sean Humphray from Illumina. We would like to acknowledge the support of The University of Cambridge, Cancer Research UK (grant numbers A11906, A20240; to N.R.), the European Research Council under the European Union's Seventh Framework Programme (FP/2007-2013)/ERC Grant Agreement no. 337905 (to N.R.), and Hutchison Whampoa Limited (to N.R.), and National Medical Research Council, Singapore (to T.M.C.).

## Author contributions

DWYT, TE, TMC and NR initiated and designed the study. DWYT, CGS, DC, TF, MM, FM, DG and NR developed methods. DWYT, MM, TMC and NR analysed the data, with assistance from OMR, CC, CGS, DC, FM, JM and WL. ASCW, RAS, HLL, BCG and TMC are the treating physicians of the patients included in this study, and collected samples and clinical data. DWYT, MM, TMC and NR interpreted the data and wrote the manuscript with assistance from other authors. All authors approved the final manuscript.

## Conflict of interest

T.F., D.G. and N.R. are co-founders, shareholders and employees/consultants of Inivata Ltd., a company that seeks to commercialize ctDNA technologies and has licensed patents and technologies from Cancer Research Technology and the University of Cambridge. D.T. and F.M are former consultants of Inivata Ltd. D.T., M.M., T.F., F.M., J.M., D.G. and N.R. may receive royalties related to the licences of IP to Inivata Ltd, and the terms of these royalties are managed by Cancer Research Technology and Cambridge Enterprise. T.E. and N.R. have received research support from AstraZeneca. T.E. is an employee of AstraZeneca on leave of absence from the University of Cambridge.

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
