## [Review Process File · EMBO Molecular Medicine]

Dynamics of multiple resistance mechanisms in plasma DNA during EGFR-targeted therapies in NSCLC

Dana Wai Yi Tsui, Muhammed Murtaza, Alvin Seng Cheong Wong, Oscar M Rueda, Chris G Smith¹, Dineika Chandrananda, Ross A Soo, Hong Liang Lim, Boon Cher Goh, Carlos Caldas, Tim Forsheew, Davina Gale, Wei Liu, James Morris, Francesco Marass, Tim Eisen, Tan Min Chin, Nitzan Rosenfeld

Review timeline:

Submission date:	25 April 2017
Editorial Decision:	10 June 2017
Revision received:	24 October 2017
Editorial Decision:	29 November 2017
Revision received:	04 April 2018
Accepted:	09 April 2018

Editor: Roberto Buccione/Céline Carret

Transaction Report:

1st Editorial Decision

10 June 2017

Thank you for the submission of your manuscript to EMBO Molecular Medicine and many apologies for the unusual delay in providing you with a decision.

We have now received comments from the two out of the three Reviewers whom we asked to evaluate your manuscript. We have been unable so far, to retrieve the third.

Hence, to avoid further delays I am sending the two consistent evaluations of Reviewers 1 and 2 at this time. I will forward Reviewer 3's delayed report, if and as soon as we are able to obtain it. When (within reason) this report does arrive and if it raises additional important issues that have to be addressed to support this study, these would also need to be taken into consideration in your revision. Please note that I would not ask you to consider further-reaching requests with respect to the current evaluations.

You will see that in aggregate, both reviewers find the study of interest, while at the same time clearly mentioning the need to clarify a number of issues. I will not dwell into much detail, as the comments are self-explanatory. However, I would like to specifically point out that reviewer 1 laments the lack of considerable technical and experimental information, which is required to fully evaluate the quality and robustness of the data, and consequently assess the solidity of the conclusions. Reviewer 2 also notes that extensive re-writing and streamlining are also required. I would also ask you to address his/her concerns on the lack of novelty, which however reviewer 1 does not raise.

In conclusion, while publication of the manuscript cannot be considered at this stage, we would be pleased to consider a suitably revised submission, provided, however, that the Reviewers' concerns are fully addressed with further experimentation where required.

Please note that it is EMBO Molecular Medicine policy to allow a single round of revision only and that, therefore, acceptance or rejection of the manuscript will depend on the completeness of your responses included in the next, final version of the manuscript.

EMBO Molecular Medicine now requires a complete author checklist (<http://embomolmed.embopress.org/authorguide#editorial3>) to be submitted with all revised manuscripts. Provision of the author checklist is mandatory at revision stage; the checklist is designed to enhance and standardize reporting of key information in research papers and to support reanalysis and repetition of experiments by the community. The list covers key information for figure panels and captions and focuses on statistics, the reporting of reagents, animal models and human subject-derived data, as well as guidance to optimise data accessibility.

We now mandate that all corresponding authors list an ORCID digital identifier. You may do so through our web platform upon submission and the procedure takes < 90 seconds to complete. We also encourage co-authors to supply an ORCID identifier, which will be linked to their name for unambiguous name identification.

Please carefully adhere to our guidelines for authors (<http://embomolmed.embopress.org/authorguide>) to accelerate manuscript processing in case of acceptance.

I look forward to seeing a revised form of your manuscript as soon as possible.

***** Reviewer's comments *****

Referee #1 (Comments on Novelty/Model System):

Tsui et al report on the analysis of ctDNA profiles for 50 stage IV NSCLC patients receiving anti-EGFR therapy using TAM-seq targeted sequencing of 6 genes and dPCR analysis for EGFR hotspot mutations. The samples from 3 patients with SCLC transformation were also analyzed by shallow WGS. Improvement of minimally-invasive diagnostics and methods for measuring therapy response and identification of resistance mechanisms are all critically important issues, especially in NSCLC. The work presented may be technically sound although some parts require clarification and there is missing information that precludes a complete technical evaluation. The present manuscript does add to the body of literature further evidence of the potential utility of ctDNA analysis in monitoring NSCLC. Concordance of EGFR mutation status between tissue and any plasma sample was high, at 95%. The impressive lead time of 6.8 months between T790M resistance mutation detection by ctDNA and clinical progression will be beneficial. In addition, this study shows the value of tracking more than EGFR mutational load in ctDNA to monitor tumor dynamics. Interestingly, patients with EGFR and TP53 mutations in ctDNA at baseline had inferior OS. This is also the first report documenting the changes in ctDNA genomic profile upon histological transformation from NSCLC to SCLC.

Referee #1 (Remarks):

Key concerns:

1. A complete evaluation of the manuscript was not possible because the Supplementary Methods was not included and some parts were inadequately described in the main text and included supplementary figures and tables.
2. For example, it is not clear why digital PCR assays were used for the three EGFR hotspots. Are these not reported adequately by TAM-seq? Additionally, since two different methods are used for MAF estimation, the source of each MAF datapoint should be made clearer. Are all del19, L858R, and T790M data in Table S5 coming from digital PCR?
3. Not enough information is provided to evaluate the adequacy of the evaluation of the digital PCR assays as shown in Table S7 and Figure S5. As shown in Table S7, why are different numbers of tests run for the 3 mutation assays, and why is there significant variation in the amounts of input

DNA per test and between the assays? The number of genomes loaded per test appears to be low compared to the average advanced NSCLC patient.

4. The definition of false-positive rate in Table S7 should be clarified so as to not be misinterpreted. For L858R, 3 out of 28 tests have a false-positive call, and the number of genomes included per test is relatively low (100-600, average 340 genomes). How does this type of false-positive rate influence the threshold used to call a patient plasma sample as positive or negative? Typically T790M is a more difficult assay than L858R, because false positives affect transitions more than transversions, therefore it is surprising that T790M has fewer false-positives. It is notable that the number of input genomes per test is significantly lower for most of the T790M tests, and one test alone accounts for about 50% of the analyzed genomes (average 130 genomes after excluding the outlier test that contained over 5000).

5. To interpret the dilution series, the starting concentration and total genomes per test for Figure S5 A/C/D should be provided. In the Figure S5 legend, it is misleading to state that the linearity of quantification in a dilution series, especially at higher concentrations, is relevant to the "sensitivity and specificity" of the assay.

6. The recent works from the Swanton (Abbosh et al, Nature. 2017 Apr 26; Jamal-Hanjani et al, N Engl J Med. 2017 Apr 26) and Diehn labs (Chabon JJ et al, Nat Commun. 2016 Jun 10;7:11815) should be incorporated.

Minor concerns and typos:

1. In the Results section, first paragraph and in the Table S1 and in Figure 1A, it should be made clearer that concordance between tissue and plasma is evaluated by counting as true any concordant status found in not only the closest baseline sample, but including any of the follow-up plasma samples. This definition inflates the concordance statistic, so it is important that it is more clearly stated for the reader.

2. It is not obvious why the TKI-naïve subgroup in the prognosis analysis was reduced from 34 patients to 21.

3. Page 4/5: presentation of the Cox p-value of 0.06 for *either* PFS or OS as written here (and calculated as separate p-values in the figure) can be easily misinterpreted and should be clarified.

4. Have the authors tested the prognostic value of using the mutant copies/ml values instead of MAFs? Or are the concentration values subject to too much technical variation coming from the plasma DNA isolation?

5. The manuscript would be easier to follow if, where possible, the relevant patient #s were provided, so that the reader can follow the information between the text, figures, and supplementary tables. Also, please mark the patient numbers in Figure 3.

6. It would be helpful to include information on the specific EGFR exon 19 deletions present in each patient, e.g. in Table S1.

7. Which exon 19 deletion(s) does the digital PCR assay detect?

8. Typo in Table S1 - only one "no" is indicated in the EGFR status agreed column when there should be two disagreements.

9. In Table S1, case 109 is missing the TP53 mutation information.

10. It appears that Table S2 is missing some of the right-most columns.

11. Page 6, bottom paragraph: "29 patients" must be a typo?

12. Figure 4 legend does not match the panel labels.

13. It is questionable whether the day 300 timepoint for patient 103 should be excluded, as it does have positive mutation detection for ex19del at 1.9%.
14. It is challenging to find the likely explanation for the dramatic increase in ex19del and T790M in patient 103 at days 189 and 217 (incorrectly described as "increased AFs in days 244 and 272" on page 7, third paragraph), which then fall again by day 244, because these data do not track the size of the lung or liver lesions. There is also a spurious (?) detection of L858R at 0.05% at day 217. Could there be other explanations than those provided? A technical issue that led to overestimated MAFs or gives false-positives?
15. I am not sure whether the data suggests "strongly" that T790M was present in L3 but not L1/L2. T790M MAF goes up 4-fold between day 272 and day 679 and it is L1/L2 which increase in size during this time period whereas L3 decreases. Also, at day 300, where L3 is largest, ex19del is detected at a two-fold increased value (1.9%) and T790M is undetected. Even though day 300 may have had a poor extraction, it should affect all mutations equally, and if the L3 tumor is increasing at this time and ex19del MAF increased here too, then one would expect T790M to also be detected if it is present and part of the resistant clone in L3. This doesn't suggest "strongly" that T790M is driving L3. T790M increases at day 783, which is 42 days after the last CT. Do you have any more recent scans for patient 103 that could shed some light? Does this patient have other lesions that could be contributing?
16. Page 7, second paragraph, typos: "On day 297" should be day 217, and "8.5%" should be 8.6%. Also, this section would be easier to follow if the day #s were provided for the events being described.
17. Page 7, third paragraph, Figure 4C should be referenced, not 4A.
18. Figure 5, heading typo: "Patient 233" should be 223. Also, in panel B, are the TP53 and PIK3CA status boxes at day 63 giving the plasma status (which the legend should seem to indicate), or the re-biopsy status? According to Table S5, there is no day 63 plasma sample but only day 0 and day 354 plasma samples indicated for this patient.
19. Figure 5: the plasma 1 CNV plot has shows rather dramatic gains and losses. Is the ctDNA content rather high, or does some processing of the WGS data accentuate the gains and losses even with low fraction of ctDNA compared to total cfDNA? Table S5 indicates 107 copies/ml of cfDNA for this plasma 1 sample, and 2.4ml plasma was used in isolation (meaning there is not that much cfDNA in total, much less than the 5-10ng input indicated in the methods). The highest MAF in this plasma 1 sample is PIK3CA at 5.8%, which suggests a moderately low ctDNA content. Is "107" a typo? What percent ctDNA content in a plasma sample is needed to get a reliable or non-flat sWGS CNV profile? Patient 218, Figure 6 plasma 2 and 3, have quite flat CNV profiles, even though this patient's plasma 2 has a TP53 mutation at 25.8%. Can you estimate ctDNA fraction from the sWGS data?
20. Figure 6: it is very difficult to see and follow the lesions of interest in the imaging scans in panel A and the top part of panel B is too small to read.
21. Supplementary Table 4: The patients who do not have samples within the relevant time periods could be omitted or moved to the bottom or indicated in some clearer way, for the "-" marks for these patients could be misinterpreted by the reader.

Referee #2 (Remarks):

Tsui and colleagues studied the dynamics of multiple oncogenic drivers and resistance mechanisms in plasma cell free DNA of 50 patients with EGFR-mutant NSCLC, during treatment with gefitinib and hydroxychloroquine. To analyze cell-free DNA, they exploited digital PCR and TAM-Seq as well as shallow whole genome sequencing.

Interestingly, these analyses were serially performed on 3 cases who underwent histological transformation to SCLC. EGFR activating mutations were identified in 95% of patients (41/43);

additional mutations including EGFR T790M, TP53, PIK3CA and PTEN were also identified and tracked longitudinally during treatment. A relevant finding is the correlation between TP53 and EGFR detected in plasma prior to treatment and worse overall survival compared to EGFR only mutant patients.

The study is well conducted, however it is mainly observational and the findings are not completely novel.

Major points:

- this work is a patched description of different cases and it is quite difficult for the reader to follow the entire story; the flow of the manuscript should be improved
- figure readability should be improved; in particular, consistency in showing treatment schedules among different cases and related info should be implemented

Minor points:

- please check figure 4d as there is a liver CT scan (day 297) while that panel concerns lung lesions only.
- digital PCR raw data are provided with number of target copies corrected by Poisson statistics; please add confidence intervals as well
- Figure 5-6 legends should be revised and figures better explained
- supplementary tables cannot be read as the font is too small; they should be provided in the original excel format

Missing referee report

19 June 2017

I have now received the missing evaluation from reviewer 3 (please see below). You will remember that in my previous decision letter I had informed you that if the missing report raised additional important issues that have to be addressed to support this study, they would also need to be taken into consideration in your revision. I had also mentioned however, that I would be asking you to consider further-reaching requests with respect to evaluations from reviewers 1 and 2.

As you will see, reviewer 3 is rather unenthusiastic has several concerns. It appears however, that they can be mostly addressed by providing important additional details of the dataset and patients and by providing better explanations/discussion on a number of issues. I would, however bring a specific point to your attention. The reviewer notes that while the SCLC angle is interesting, the lack of consideration of RB loss makes it of uncertain significance.

I would therefore invite you to deal with the above concerns, in addition to those of the other reviewers, by providing a full point-by-point rebuttal and by introducing appropriate textual amendments in the manuscript. Although I am not specifically asking you to provide further experimentation to address them, I would encourage you to provide additional supporting data if available, especially to address the concern on SCLC.

Reviewer #3

Comments on Novelty/Model System:

This is a mixed population of patients and the analyses are not very clear.

Remarks:

The manuscript by Tsui and colleagues evaluates serial plasma DNA from patients undergoing therapy with gefitinib/hydorxylroquine. The authors perform concordance studies with tumor

assessments and analyze plasma DNA serially using Tam-Seq and shallow whole genome sequencing. The authors evaluate the emergence of resistance mechanisms.

1.) The clinical cohort is a bit of a potpourri of patients with some having EGFR mutations (and not all are del 19 or L858R) and others being EGFR WT while a subset have developed resistance to prior EGFR inhibitor treatment. It would be useful to have the clinical data in the manuscript as it is hard to follow without it. What was the response rate, PFS etc. for the patients (as a whole and then in the two cohorts) ?

2.) Figure 1A. The authors should comment on finding two EGFR mutations in the plasma (Del 19 and L858R) of two patients. Are both mutations found in the tumor ? Are the L858R mutations false positives ? Are these ones with low AF ? The authors should verify these using an orthogonal method (like ddPCR). Minor point - the first row on Figure 1A (exon 19 deletion) - the numbers do not add up to 23 (they add up to 25).

3.) What are the clinical characteristics of patients with "low" levels of baseline plasma detected EGFR mutations ? Are these patients with low volume disease to begin with or chest only stage IV NSCLC etc.

4.) The mutation dynamics for T790M detection is confusing - The authors mention that 28/45 patients developed T790M. However, the cohort of patients that are TKI naïve is only 34 patients. The analysis seems to combine both treatment naïve and treated cohorts which is confusing since some of the patients are already resistant. The authors should focus here solely on the TKI cohort and report: a.) in how many patients did they detect T790M prior to clinical progression and b.) what was the median time (and range) from detecting the T790M mutation in plasma to the time the patient experienced clinical progression.

5.) The authors should be a bit more speculative in their discussion of the "third-group" of patients (page 6). There are no examples to date of patients with metastatic EGFR mutant (tumor genotyped) lung cancer patients treated with an EGFR inhibitor who lose their EGFR mutation. EGFR mutations are truncal events. The author's conclusion can only be made using tumor genotyping which should be presented. The findings are otherwise only speculative. The authors should also present the tumor genotype of these 7 patients (since it was known for the majority of patients pre-treatment on the study).

6.) The SCLC findings are interesting but there is no discussion or mention of RB. All SCLC (and EGFR mutant SCLCs) have mutations (or loss) in TP53 and RB. Also all EGFR mutant SCLC retain their EGFR mutation which is not the case in 1 of the patients here. Did the authors sequence the SCLC biopsies ? SCLC is a histological diagnosis not a molecular diagnosis. The authors need to discuss this limitation in their study - no one would treat a patient with SCLC directed therapy based on cfDNA profiling alone. The only clinically actionable plasma DNA directed genotype in EGFR mutant patients who develop resistance to EGFR TKIs is T790M. This point needs to also be articulated in the discussion.

1st Revision - authors' response

24 October 2017

Referee #1 (Comments on Novelty/Model System):

Tsui et al report on the analysis of ctDNA profiles for 50 stage IV NSCLC patients receiving anti-EGFR therapy using TAM-seq targeted sequencing of 6 genes and dPCR analysis for EGFR hotspot mutations. The samples from 3 patients with SCLC transformation were also analyzed by shallow WGS. Improvement of minimally-invasive diagnostics and methods for measuring therapy response and identification of resistance mechanisms are all critically important issues, especially in NSCLC. The work presented may be technically sound although some parts require clarification and there is missing information that precludes a complete technical evaluation. The present manuscript does add to the body of literature further evidence of the potential utility of ctDNA analysis in monitoring NSCLC. Concordance of EGFR mutation status between tissue and any plasma sample was high, at 95%. The impressive lead time of 6.8 months between T790M resistance mutation detection by ctDNA and clinical progression will be beneficial. In addition, this study shows the value of tracking

more than EGFR mutational load in ctDNA to monitor tumor dynamics. Interestingly, patients with EGFR and TP53 mutations in ctDNA at baseline had inferior OS. This is also the first report documenting the changes in ctDNA genomic profile upon histological transformation from NSCLC to SCLC.

We thank Referee #1 for acknowledging the important clinical relevance of our work.

Referee #1 (Remarks):

We thank referee #1 for providing their constructive feedback, please see our responses outlined below:

Key concerns:

1. A complete evaluation of the manuscript was not possible because the Supplementary Methods was not included and some parts were inadequately described in the main text and included supplementary figures and tables.

We sincerely apologize for the error. The Supplementary Methods and legends for the Appendix Tables are now provided in Appendix. We also indicated clearly in the main text referral to the Appendix Materials.

2. For example, it is not clear why digital PCR assays were used for the three EGFR hotspots. Are these not reported adequately by TAM-seq? Additionally, since two different methods are used for MAF estimation, the source of each MAF datapoint should be made clearer. Are all del19, L858R, and T790M data in Table S5 coming from digital PCR?

Digital PCR is more sensitive than TAM-Seq and offers a theoretical detection limit of 1 in 1000 molecules (0.1%). Given the clinical relevance of EGFR hotspot mutations (L858R, Ex19 15-bp deletion, and EGFR T790M), we applied both technologies to quantify MAF and reported the average values obtained using the two technologies in Table S5. We added a sentence to supplementary methods (Page 3, last sentence) to clarify the calculation.

3. Not enough information is provided to evaluate the adequacy of the evaluation of the digital PCR assays as shown in Table S7 and Appendix Fig S5. As shown in Table S7, why are different numbers of tests run for the 3 mutation assays, and why is there significant variation in the amounts of input DNA per test and between the assays? The number of genomes loaded per test appears to be low compared to the average advanced NSCLC patient.

To establish specificity of the assays, we aimed to test around 10,000 presumably wild-type molecules to check if there are any false positive signals. We did that by repeating multiple analyses using a large amount of control samples and sum up all the poisson-corrected counts of wild-type DNA. The number of reaction chambers available in each well of the Fluidigm BioMark chip is 765. We therefore deliberately diluted the input materials to not exceed that number in most tests. Therefore the numbers of wild-type molecules in each replicated test runs vary and different numbers of tests run (or replicates) were required to achieve ~10,000 total number of wild-type molecules.

Regarding the point about the low number of genomes loaded per test, we apologize for the lack of clarification: the data of the multiple test runs were actually generated using a large amount of pooled healthy individual plasma samples. The multiple test runs should therefore be considered technical replicates. The reasons why multiple replicates were required are, like above, because of the limited number of chamber in each well and the need to gather ~10,000 total molecules. If this experiment is to be performed in a different dPCR platform which allows higher levels of compartmentation (such as droplet-based technology), then multiple replicates will not be necessary. This piece of information is now added to supplementary materials Page 3, second paragraph.)

4. The definition of false-positive rate in Table S7 should be clarified so as to not be misinterpreted. For L858R, 3 out of 28 tests have a false-positive call, and the number of genomes included per test is relatively low (100-600, average 340 genomes). How does this type of false-positive rate influence the threshold used to call a patient plasma sample as positive or negative? Typically

T790M is a more difficult assay than L858R, because false positives affect transitions more than transversions, therefore it is surprising that T790M has fewer false-positives. It is notable that the number of input genomes per test is significantly lower for most of the T790M tests, and one test alone accounts for about 50% of the analyzed genomes (average 130 genomes after excluding the outlier test that contained over 5000).

As outlined in our answer to comment 3 above, the data in the multiple rows per assay were technical replicates generated using a large amount of pooled healthy individual plasma samples (details please see answer 3). Therefore, the definition of false-positive rate in Table S7 is given by the number of expected count of mutant DNA over the total sum of wild-type DNA in each assay. For example for L858R, the numbers of mutant DNA detected are 3 out of a total of 9625 wild-type molecules and hence the false positive rate is 0.03%. Again, we apologize for this confusion.

5. To interpret the dilution series, the starting concentration and total genomes per test for Appendix Fig S5 A/C/D should be provided. In the Appendix Fig S5 legend, it is misleading to state that the linearity of quantification in a dilution series, especially at higher concentrations, is relevant to the "sensitivity and specificity" of the assay.

The numbers of genomes per test input to the test were 1-9 copies of mutant DNA spiked in to 748-955 copies of wild-type DNA (please see Appendix Methods Page 3). Again, we apologize for missing the Appendix Methods document in the initial submission. We took the referee's advice and modified the language from "Sensitivity and specificity" to "linearity of quantification" in Appendix Fig S5 legend (Appendix Figures Page 5).

6. The recent works from the Swanton (Abbosh et al, Nature. 2017 Apr 26; Jamal-Hanjani et al, N Engl J Med. 2017 Apr 26) and Diehn labs (Chabon JJ et al, Nat Commun. 2016 Jun 10;7:11815) should be incorporated.

We have added the suggested literature as reference #20 (Page 3), #31 (Page 11) and #33 (Page 12), respectively.

Minor concerns and typos:

1. In the Results section, first paragraph and in the Table S1 and in Figure 1A, it should be made clearer that concordance between tissue and plasma is evaluated by counting as true any concordant status found in not only the closest baseline sample, but including any of the follow-up plasma samples. This definition inflates the concordance statistic, so it is important that it is more clearly stated for the reader.

We have clarified in the first paragraph of the Results section (Page 4) that any follow-up plasma samples were being considered in the concordance analysis.

2. It is not obvious why the TKI-naïve subgroup in the prognosis analysis was reduced from 34 patients to 21.

The 21 patients were selected for the analysis because they have at least 1 plasma sample collected before treatment was initiated. We have now clarified that in the second paragraph of the results section (Page 5).

3. Page 4/5: presentation of the Cox p-value of 0.06 for *either* PFS or OS as written here (and calculated as separate p-values in the figure) can be easily misinterpreted and should be clarified.

We have now clarified the Cox p-values are for both PFS and OS and make a clear distinction from the p-values shown in the figure (Page 5).

4. Have the authors tested the prognostic value of using the mutant copies/ml values instead of MAFs? Or are the concentration values subject to too much technical variation coming from the plasma DNA isolation?

We tried the analysis using mutant copies/ml instead of MAFs and the conclusions are the same (page 5)

5. The manuscript would be easier to follow if, where possible, the relevant patient #s were provided, so that the reader can follow the information between the text, figures, and supplementary tables. Also, please mark the patient numbers in Figure 3.

We thank the reviewer for the suggestion and have added the relevant patient number to figure 3 and the legend for the readers to follow the information.

6. It would be helpful to include information on the specific EGFR exon 19 deletions present in each patient, e.g. in Table S1.

The requested information has been added to Table S1.

7. Which exon 19 deletion(s) does the digital PCR assay detect?

The dPCR assay was deliberately designed to detect several different types of Ex19 deletion (that span 15-18 bp from amino acid 745 to 759) (Details please see Yung et al 2009 Clin Chem)

8. Typo in Table S1 - only one "no" is indicated in the EGFR status agreed column when there should be two disagreements.

We apologize for this error and have corrected Table S1 and changed "yes" to "no" in well D46 to accurately reflect the data of patient 131.

9. In Table S1, case 109 is missing the TP53 mutation information.

The requested information has been added to Table S5.

10. It appears that Table S2 is missing some of the right-most columns.

This error was due to a page break, we apologize. We have replaced with an updated complete version of Table S2.

11. Page 6, bottom paragraph: "29 patients" must be a typo?

We thank the reviewer for indicating this. We have corrected the typo to "13 patients" (Now page 7).

12. Figure 4 legend does not match the panel labels.

We have improved the clarity of the panel labels by adding (i) to (iii) sub-panels under each panel A and B.

13. It is questionable whether the day 300 timepoint for patient 103 should be excluded, as it does have positive mutation detection for ex19del at 1.9%.

The sample collected on day 300 from patient 103 presented an unexpectedly low total cfDNA level (>10-fold different from the timepoint immediately before and after), which could potentially influence the interpretation of mutant allele fractions at that time-point. Such variations could be contributed by effects of processing, collection or other technical reasons. We have therefore excluded that timepoint from the analysis. We have clarified this point in the results section (page 8)

14. It is challenging to find the likely explanation for the dramatic increase in ex19del and T790M in patient 103 at days 189 and 217 (incorrectly described as "increased AFs in days 244 and 272" on page 7, third paragraph), which then fall again by day 244, because these data do not track the size of the lung or liver lesions. There is also a spurious (?) detection of L858R at 0.05% at day 217. Could there be other explanations than those provided? A technical issue that led to overestimated MAFs or gives false-positives?

We thank the reviewer for pointing this out. Regarding the statement "increased AFs in days 244 and 272", we apologized for the confusion and have corrected that to "increased AFs from days 244 to 272" (Page 8, second paragraph), which more truly reflected the data. Regarding the potential technical issues, as described in the text, from day 217 to day 244, both EGFR mutations (activating and resistant) exhibited a sharp drop in AF in plasma for reasons we do not understand. Similarly, on day 300 there was a 10-fold drop in total cfDNA level. These could be due to pre-analytical technical reasons such as sub-optimal sample processing protocols, which has been proven to be an important factor influencing the levels of total cfDNA and corresponding tumor-derived allele fractions [Parpart-Li et al Clin Cancer Res 2016]. Regarding the detection of 0.05% L858R at day 217, it is above the levels of false positive rate of our digital PCR assay (which was estimated to be 0.03%, see Appendix Methods). The fact that it was only detected in one out of the 10 plasma samples suggested that it might be a transient appearance or a sub-clonal event. We therefore attempted to normalize for such effects by studying the relative representation of the sensitizing and resistance-conferring alleles, which should theoretically be influenced to a similar extent by the pre-analytical factors, if any.

15. I am not sure whether the data suggests "strongly" that T790M was present in L3 but not L1/L2. T790M MAF goes up 4-fold between day 272 and day 679 and it is L1/L2 which increase in size during this time period whereas L3 decreases. Also, at day 300, where L3 is largest, ex19del is detected at a two-fold increased value (1.9%) and T790M is undetected. Even though day 300 may have had a poor extraction, it should affect all mutations equally, and if the L3 tumor is increasing at this time and ex19del MAF increased here too, then one would expect T790M to also be detected if it is present and part of the resistant clone in L3. This doesn't suggest "strongly" that T790M is driving L3. T790M increases at day 783, which is 42 days after the last CT. Do you have any more recent scans for patient 103 that could shed some light? Does this patient have other lesions that could be contributing?

We thank the reviewer for point this out and have revised the language from "suggest strongly" to "...suggests that the T790M may possibly be present in L3...". Unfortunately this patient had since passed on, and no further scans are available

16. Page 7, second paragraph, typos: "On day 297" should be day 217, and "8.5%" should be 8.6%. Also, this section would be easier to follow if the day #s were provided for the events being described.

Day 297 is accurate; the dates of tumor measurement and blood collection do not always fall on the same day. "8.5%" was a typo and we have revised to "8.6%" the information and added the day number to indicate more clearly the timeline of the events.

17. Page 7, third paragraph, Figure 4C should be referenced, not 4A.

We have corrected the figure number accordingly.

18. Figure 5, heading typo: "Patient 233" should be 223. Also, in panel B, are the TP53 and PIK3CA status boxes at day 63 giving the plasma status (which the legend should seem to indicate), or the re-biopsy status? According to Table S5, there is no day 63 plasma sample but only day 0 and day 354 plasma samples indicated for this patient.

We thank the reviewer for pointing this out and we have corrected the typo. Day 63 refers to the date of collection of the re-biopsy tumor only. No plasma samples were collected on that day.

19. Figure 5: the plasma 1 CNV plot has shows rather dramatic gains and losses. Is the ctDNA content rather high, or does some processing of the WGS data accentuate the gains and losses even with low fraction of ctDNA compared to total cfDNA? Table S5 indicates 107 copies/ml of cfDNA for this plasma 1 sample, and 2.4ml plasma was used in isolation (meaning there is not that much cfDNA in total, much less than the 5-10ng input indicated in the methods). The highest MAF in this plasma 1 sample is PIK3CA at 5.8%, which suggests a moderately low ctDNA content. Is "107" a typo? What percent ctDNA content in a plasma sample is needed to get a reliable or non-flat sWGS CNV profile? Patient 218, Figure 6 plasma 2 and 3, have quite flat CNV profiles, even though this

patient's plasma 2 has a TP53 mutation at 25.8%. Can you estimate ctDNA fraction from the sWGS data?

We acknowledge and thank the reviewer for pointing out the clarification which might be needed for the plasma 1 copy number plot. We have added text to clarify that the overall read count was lower for this sample as compared to the other plots shown. This has the effect of increasing the apparent noise levels of the bins (I.e. each dot in the plot) which in turn accentuates the apparent copy number gains and losses. Importantly, the segmentation algorithm we employ produces a copy number landscape (as indicated by orange horizontal lines) that resembles those obtained in the matched longitudinal samples, suggesting that the added noise in this sample is not masking the true copy number landscape.

It has been found, through the use of serial dilution experiments, that the sensitivity of shallow whole genome sequencing for detection of copy-number abnormal ctDNA in plasma is ~10% (Heitzer et al, 2013 Genome Medicine). Whilst we agree that it can be useful to compare ctDNA levels as inferred by the amplitude of copy number abnormalities and SNV allele fractions, there is still much we do not know about the interaction of these variables, not to mention the role that differing clonal dynamics have within a given disease. Thus, whilst a plasma sample at a given time point might have a somatic SNV with a reasonably high AF of 30%, it does not necessarily mean that the copy number profile at that time point will have many aberrations e.g. It may be that the dominant clone(s) containing the 30% AF SNV contains less copy number aberrations as compared to the other clones which make up the disease, and vice-versa.

20. Figure 6: it is very difficult to see and follow the lesions of interest in the imaging scans in panel A and the top part of panel B is too small to read.

We have modified the figures and legends accordingly to clarify.

21. Appendix Table S4: The patients who do not have samples within the relevant time periods could be omitted or moved to the bottom or indicated in some clearer way, for the "-" marks for these patients could be misinterpreted by the reader.

We have modified Table S4 as suggested.

Referee #2 (Remarks):

Tsui and colleagues studied the dynamics of multiple oncogenic drivers and resistance mechanisms in plasma cell free DNA of 50 patients with EGFR-mutant NSCLC, during treatment with gefitinib and hydroxychloroquine. To analyze cell-free DNA, they exploited digital PCR and TAM-Seq as well as shallow whole genome sequencing. Interestingly, these analyses were serially performed on 3 cases who underwent histological transformation to SCLC. EGFR activating mutations were identified in 95% of patients (41/43); additional mutations including EGFR T790M, TP53, PIK3CA and PTEN were also identified and tracked longitudinally during treatment. A relevant finding is the correlation between TP53 and EGFR detected in plasma prior to treatment and worse overall survival compared to EGFR only mutant patients. The study is well conducted, however it is mainly observational and the findings are not completely novel.

We thank the reviewer for acknowledging the quality of our study. To our knowledge, this is the first study to longitudinally track the dynamics of molecular profiles of multiple NSCLC patients from the initiation of EGFR-targeted therapy to histological transformation to SCLC and follow their therapy responses by cell-free DNA analysis. A recent study reported that EGFR TKI-resistance SCLCs branched out from early events that pre-existed in NSCLC prior to transformation based on tumor biopsy analysis [Lee et al 2017 JCO]. However, repeated biopsies are practically not feasible and our findings suggest that it is possible to non-invasively track such molecular dynamics by plasma cell-free DNA.

Major points:

1) this work is a patched description of different cases and it is quite difficult for the reader to follow the entire story; the flow of the manuscript should be improved

To address the reviewer's concern about the flow of the manuscript and the clarification about the differences in treatment among different cases, we have now included a consort figure to guide the readers to follow the number of cases involved in each component of the study such as concordance between tumor and plasma and survival analysis.

2) figure readability should be improved; in particular, consistency in showing treatment schedules among different cases and related info should be implemented

We have modified the font and image sizes of the figures to improve readability and consistency of the presentation of treatment schedules.

Minor points:

1) please check figure 4d as there is a liver CT scan (day 297) while that panel concerns lung lesions only.

We have revised the figures accordingly.

2) digital PCR raw data are provided with number of target copies corrected by Poisson statistics; please add confidence intervals as well

We have added the confidence intervals to Table S5.

3) Figure 5-6 legends should be revised and figures better explained. Appendix tables cannot be read as the font is too small; they should be provided in the original excel format

We have revised the said figure legends and improved the readability of the supplementary tables. Unfortunately the submission system does not allow us to provide the excel file and therefore we provided it in Word format.

Reviewer #3

Comments on Novelty/Model System:

This is a mixed population of patients and the analyses are not very clear.

Remarks:

The manuscript by Tsui and colleagues evaluates serial plasma DNA from patients undergoing therapy with gefitinib/hydroxychloroquine. The authors perform concordance studies with tumor assessments and analyze plasma DNA serially using Tam-Seq and shallow whole genome sequencing. The authors evaluate the emergence of resistance mechanisms.

1.) The clinical cohort is a bit of a potpourri of patients with some having EGFR mutations (and not all are del 19 or L858R) and others being EGFR WT while a subset have developed resistance to prior EGFR inhibitor treatment. It would be useful to have the clinical data in the manuscript as it is hard to follow without it. What was the response rate, PFS etc. for the patients (as a whole and then in the two cohorts)?

This is a phase II with a lead-in phase I study to study the tolerability, safety profile and efficacy of hydroxychloroquine and gefitinib in advanced non-small cell lung cancer. The clinical manuscript is in preparation, and will report the response rate, progression free and overall survival of the two different cohorts of patients. For the purpose of allowing the reviewer to better understand the context, below is a brief description of the clinical study:

In the phase I part of the study, 13 patients were recruited to study safety and tolerability of 600 mg of gefitinib daily in combination with 250 mg of hydroxychloroquine daily. At the time of recruitment, EGFR wild type patients were eligible for the phase I part of the study (n=2), although most patients were enriched by either clinical or molecular selection. All patients included in the phase II part of this study in the EGFR TKI naïve group were known EGFR mutation carriers by tumor molecular typing. In the EGFR TKI treated group, most were eligible either by known sensitising mutations in the tumor, or a history of prior disease control for at least 12 weeks to previous EGFR TKI.

A consort diagram has been included to present the number of patients in each arm.

2.) Figure 1A. The authors should comment on finding two EGFR mutations in the plasma (Del 19 and L858R) of two patients. Are both mutations found in the tumor? Are the L858R mutations false positives? Are these ones with low AF? The authors should verify these using an orthogonal method (like ddPCR). Minor point - the first row on Figure 1A (exon 19 deletion) - the numbers do not add up to 23 (they add up to 25).

We acknowledge this important comment by the reviewer. Of the 23 patients whose tumor was found to harbour EGFR Exon 19 deletions, the same deletions were detected in the plasma samples of 21 patients, of which, 2 patients also had L858R mutation detected in the same plasma sample. These mutations were detected above the false positive noise level of our assays (See Appendix Methods.) Unfortunately the original tumor biopsy materials of these two patients have been depleted and therefore we were unable to verify whether both EGFR mutations were present in the tumors. Other studies have reported the co-existence of Exon 19 deletions and L858R mutations in plasma samples of NSCLC patients (e.g. Couraud et al 2014 Clin Can Res, Paweletz et al 2015 Clin Can Res), which highlights the ability of plasma to capture tumor heterogeneity. It is worth noting that in the current study the L858R mutations were both detected at <1% allele fraction in plasma, and they may represent a sub-clonal event that co-exist with the main driver Exon 19 deletion. Regarding the numbers shown on Figure 1A, the 21 patients that had Exon 19 deletions detected in plasma overlap with the 2 patient who also had L858R mutation, and hence the numbers appears exceeding the total but they are accurate. We apologize for the confusion, and have now modified the figure to clarify.

3.) What are the clinical characteristics of patients with "low" levels of baseline plasma detected EGFR mutations? Are these patients with low volume disease to begin with or chest only stage IV NSCLC etc.

We have now included the tumor measurements (RECIST) of patients with low versus high levels of baseline plasma detected EGFR mutations to Appendix Table S3. Although the clinical characteristics of the three groups of patients were not different, (majority had chest only disease), and only one patient per group had extra-thoracic disease, the tumor burden via RECIST was higher in the group with higher mutation titres. This would suggest that plasma mutant titres are a good surrogate for disease burden, and may explain for its correlation with prognosis of patients.

4.) The mutation dynamics for T790M detection is confusing - The authors mention that 28/45 patients developed T790M. However, the cohort of patients that are TKI naïve is only 34 patients. The analysis seems to combine both treatment naïve and treated cohorts which is confusing since some of the patients are already resistant. The authors should focus here solely on the TKI cohort and report: a.) in how many patients did they detect T790M prior to clinical progression and b.) what was the median time (and range) from detecting the T790M mutation in plasma to the time the patient experienced clinical progression.

We now have added a consort diagram showing the number of patients analysed for T790M in the EGFR TKI naïve and treated arms respectively. In the 34 EGFR TKI naïve patients, 16 had T790M, giving a positivity rate of 52%. In the 13 EGFR TKI treated patients, 8 had T790M, giving a positivity rate of 62%.

5.) The authors should be a bit more speculative in their discussion of the "third-group" of patients (page 6). There are no examples to date of patients with metastatic EGFR mutant (tumor genotyped) lung cancer patients treated with an EGFR inhibitor who lose their EGFR mutation. EGFR mutations are truncal events. The author's conclusion can only be made using tumor genotyping which should be presented. The findings are otherwise only speculative. The authors should also present the tumor genotype of these 7 patients (since it was known for the majority of patients pre-treatment on the study).

We agree with the reviewer that our findings challenge the conventional understanding about EGFR mutations in NSCLC as founder events. However, recent study performed based on tumor biopsy analysis has also suggested that it is possible EGFR may be sub-clonal in 14% (3/21) of NSCLC (McGranahan et al Sci Transl Med 2015). In the current study, 15% (7/45) patients show decreasing

EGFR activating mutations levels in plasma despite progression, at the same time, we also identified other potential drivers TP53 or PIK3CA at higher MAF in some of the first plasma samples, suggesting the possibility that these cancers may be driven by pathways other than EGFR. Similar findings have also been recently reported in Pecuchet et al PloS Medicine 2016, in which the authors found 4 out of 24 patients had EGFR sensitizing mutations detected in plasma at T0 but absent when the patients progressed. These findings agreed with our results.

The 7 patients initially had Exon 19 deletion detected in the tumor (7/7) and their first plasma sample (6/7). Interestingly, comparing to the other two groups, this group of patients had EGFR activating mutations present at relatively lower allele fractions in their first plasma samples (group 1 and 2: median EGFR mutations MAF was 3% (range: 0.07% – 65.7% versus group 3: median 0.23% (range: 0.06%-2.11%). We do not rule out the possibility that the tumors of these patients might release less tumor-derived DNA into the circulation. We have now added this possibility to the results and discussion.

6.) The SCLC findings are interesting but there is no discussion or mention of RB. All SCLC (and EGFR mutant SCLCs) have mutations (or loss) in TP53 and RB. Also all EGFR mutant SCLC retain their EGFR mutation which is not the case in 1 of the patients here. Did the authors sequence the SCLC biopsies? SCLC is a histological diagnosis not a molecular diagnosis. The authors need to discuss this limitation in their study - no one would treat a patient with SCLC directed therapy based on cfDNA profiling alone. The only clinically actionable plasma DNA directed genotype in EGFR mutant patients who develop resistance to EGFR TKIs is T790M. This point needs to also be articulated in the discussion.

We agree with the reviewer that it is important to study if there is any evidence of RB1 aberrations after the histological transformation to SCLC. Our targeted sequencing panel unfortunately did not include RB1 gene for mutational analysis. However, we have performed shallow whole genome sequencing using all the plasma samples and available tumor SCLC biopsy of the three SCLC-transformed patients. We did not find any evidence of significant copy number aberrations. It is worth noting that, in patient 218, RB1 does appear to have a reduced copy number, but the data also suggest that this loss could be part of a much larger chromosomal aberration and thus we cannot rule this out as being a passenger event. Indeed, RB1 alterations are important driver in SCLC but tumor lacking RB1 mutations has also been documented (Karachaliou et al 2016 Transl Lung Cancer Res). All three SCLC-transformed patients have evidence of TP53 mutations in their SCLC re-biopsies and pre- and post-transformation plasma suggest that TP53 is an important driver in these particular patients. We have now added to the results the description of the attempt to identify RB1 alterations and our findings.

We also agree with the reviewer that the molecular evidence in plasma DNA alone is not sufficient to conclude a SCLC diagnosis. However the ability to reveal TP53 mutations or other possible SCLC-associated genomic signatures in plasma would provide additional insight into possible resistance mechanisms that are particularly important in individuals that show no evidence of T790M or other known resistance mechanisms, and may justify the need for a re-biopsy to confirm the histological transformation. We have now added this point to our discussion and, as the reviewer suggested, clarified that EGFR T790M remains the only clinically actionable alterations in this context.

Thank you for the submission of your revised manuscript to EMBO Molecular Medicine. We have now received the enclosed reports from the referees that were asked to re-assess it. As you will see the reviewers are now globally supportive and I am pleased to inform you that we will be able to accept your manuscript pending final editorial amendments.

Referee #1 (Remarks for Author):

I thank the authors for revising their manuscript and for now providing the Supplementary Methods for review. The authors have adequately addressed my previous comments and concerns and I have no new remarks.

Referee #3 (Remarks for Author):

The authors have addressed the queries from this reviewer. The revised manuscript is clearer.

2nd Revision - authors' response

04 April 2018

We appreciate the very constructive feedback and recommendations provided by the Editors. We have carefully revised our manuscript. Please find the revision enclosed.

Corresponding Author Name: WY Tsui
Journal Submitted to: EMBO Molecular Medicine
Manuscript Number: EMM-2017-07945